# Dual parallel stream-specific and generalized effects of corticogeniculate feedback on LGN neurons in primate and carnivore

Sabrina Mai[1,2,3,4], Allison J. Murphy [1,2,3,4,5], J. Michael Hasse[6] &
Farran Briggs [1,2,3,4,7,8] ✉

Sensory circuits are organized in parallel, e.g. parallel streams relay feedforward visual information from retina to cortex. Corticogeniculate (CG) feedback is also organized in parallel; however, stream-specific influences of CG feedback remained unresolved. We utilized optogenetics to manipulate CG feedback in monkeys while recording geniculate responses to a comprehensive set of visual stimuli designed to probe stream-specific responses. Here we show that CG feedback improved the spatial resolution of magnocellular, but not parvocellular neurons. Optogenetically enhancing CG feedback increased extraclassical surround suppression, shrunk classical receptive fields, and increased preferred spatial frequencies among magnocellular neurons. Optogenetically suppressing CG feedback reduced surround suppression. Enhancing CG feedback in female ferrets revealed similar stream-specific effects in geniculate Y, but not X neurons. Furthermore, optogenetically enhancing CG feedback improved temporal response precision across neuronal types. These results support dual functional roles for CG feedback in enhancing spatial resolution in a stream-specific manner and improving temporal precision broadly.

A defining feature of sensory circuitry is parallel organization. A prime example of parallel sensory circuit organization is the early visual system in highly visual mammals, such as primates and carnivores. Feedforward circuits linking the primate retina to the dorsal lateral geniculate nucleus of the thalamus (LGN) to primary visual cortex (V1) are anatomically and physiologically segregated into parallel parvocellular, magnocellular, and koniocellular streams that each convey unique information about the visual world[1]. Importantly, the first feedback circuit in the visual processing hierarchy, made up of corticogeniculate (CG) neurons in V1 that send axons to the LGN, is also organized into anatomically and physiologically distinct parallel streams that align with the feedforward parvocellular, magnocellular,

and koniocellular streams[2]. However, whether CG feedback is functionally parallel-stream-specific is not clear. Mixed synaptic connectivity by CG axons onto diverse LGN cell types[3–6] could provide functionally stream-mixed feedback. Manipulations of CG feedback across species reveal inconsistent gain modulations and global modulations of response variability, with minimal effects specific to particular cell types, reviewed in ref. 2, suggesting CG feedback is a generalized modulator. In stark contrast, targeted lesions of the parvocellular or magnocellular layers of the LGN cause distinct visual perceptual deficits[7,8]. So, while feedforward visual circuits are functionally stream-specific, a significant open question remains whether CG feedback in any species exerts stream-specific influence over LGN

[1]Neuroscience Graduate Program, University of Rochester, Rochester, NY 14642, USA. [2]Department of Neuroscience, University of Rochester School of Medicine, Rochester, NY 14642, USA. [3]Ernest J. Del Monte Institute for Neuroscience, University of Rochester School of Medicine, Rochester, NY 14642, USA. [4]Center for Visual Science, University of Rochester, Rochester, NY 14642, USA. [5]Zanvyl Krieger Mind/Brain Institute, Johns Hopkins University, Baltimore, MD 21218, USA. [6]Center for Neural Science, New York University, New York, NY 10003, USA. [7]Department of Brain and Cognitive Sciences, University of Rochester, Rochester, NY 14642, USA. [8]Laboratory of Sensorimotor Research, National Eye Institute, NIH, Bethesda, MD 20892, USA. ✉e-mail: farran.briggs@nih.gov

activity. There are two alternatives: CG feedback exerts stream-specific effects that have yet to be measured, or CG feedback is functionally homogeneous albeit anatomically separated and physiologically distinct. Given that neurons in the feedforward streams have dramatically different responses to, for example, visual stimuli that encroach beyond the classical receptive field into the extraclassical surround, whether CG feedback exerts stream-specific effects or not has significant implications for the functional contributions of CG feedback to fundamental computations like extraclassical surround suppression. Until recently, selective and reversible causal manipulation of CG feedback in highly visual mammals, like primates, was not possible. Using virus-mediated circuit tracing and optogenetics, we directly tested whether CG feedback exerts stream-specific effects on LGN neurons in primates and carnivores, including exploring functional contributions of CG feedback to surround suppression.

The functional contributions of CG feedback to visual perception have been particularly challenging to parse because CG influence over LGN neurons is modulatory compared to the driving inputs LGN neurons receive from the retina[9]. For example, primate LGN neurons have circular (center-surround) receptive fields that reflect their retinal inputs, not the inputs they receive from orientation-tuned CG neurons. However, CG neurons have tuning preferences that are aligned with their LGN targets. Specifically, parvocellular LGN neurons have small, L/M-cone opponent receptive fields, prefer higher spatial frequencies at a given eccentricity, have linear responses to increasing stimulus contrast, prefer lower temporal frequencies, and display minimal surround suppression[10–13]. Likewise, CG neurons targeting parvocellular LGN layers are L/M-cone modulated, have linear responses to increasing stimulus contrast, prefer lower temporal frequencies, and have minimal surround suppression[14]. Magnocellular LGN neurons have larger, achromatic receptive fields, prefer lower spatial frequencies, have non-linear contrast response curves, follow higher temporal frequencies, and are strongly suppressed by stimuli that encroach into the extraclassical surround[12,13,15,16]. CG neurons targeting magnocellular LGN layers are achromatic, have non-linear contrast curves, prefer higher temporal frequencies, and display extraclassical surround suppression[14]. Koniocellular LGN neurons often display tuning preferences intermediate to those of parvocellular and magnocellular neurons, but are noted for carrying S-cone signals[17,18]. CG neurons thought to target koniocellular LGN neurons are also S-cone modulated and display tuning preferences intermediate to those of other CG types[14]. What is the purpose of this precise alignment of physiological preferences between feedforward and feedback geniculo-cortico-geniculate circuits if CG feedback has no stream-specific functional contribution?

A handful of studies have attempted to address this question in primates and other highly visual mammals such as cats and ferrets. Approaches used previously to manipulate CG feedback varied in severity, from non-selective disruptions of the cortex (lesion, cooling, muscimol injection) to more targeted optogenetic manipulations, likely contributing to inconsistencies across studies. For example, changes in response gain in parvocellular or magnocellular neurons (and somewhat though not strictly homologous X or Y neurons in carnivores) were observed for stimuli varying in contrast, luminance, or motion speed[19–23], but these effects were not always consistent within neuronal types and often only a limited set of stimuli were tested per study. Other effects of manipulating CG feedback, such as changes in response variability, temporal precision, or information content, were generalized across LGN neuron types, or neuron type was not considered[22,24,25]. Results of similar manipulations of corticothalamic neurons in mice suggest that feedback does not systematically alter tuning or response gain (until LGN neuronal receptive fields are distant from those of modulated CG neurons) and reduces sparseness generally among LGN neurons[26–28]. Thus, studies to date across species point to subtle effects of CG feedback on LGN neurons

with some generalized improvements in temporal precision of responses and a variety of effects on LGN neurons' spatial receptive fields or response gain not necessarily correlated with cell type. Together, these findings suggest that CG feedback is functionally homogeneous, not stream-specific.

Testing whether CG feedback is functionally homogeneous or stream-specific requires a method to probe stream-specific activity. Fortunately, because the magnocellular, parvocellular, and koniocellular streams evolved to encode different qualities of incoming visual information, it is possible to design visual stimuli that evoke distinguishable responses from neurons in each stream. Ideally, a comprehensive set of visual stimuli are utilized to maximize the uniqueness of responses across LGN neuronal types (e.g. gratings varying systematically in contrast, spatial and temporal frequency, and size in addition to measures of cone-opponency, etc.). Stimuli varying in size that engage neuronal extraclassical surrounds are especially useful for this purpose because magnocellular (and some koniocellular) LGN neurons display strong extraclassical surround suppression while most parvocellular neurons do not[13,16,29,30]. Importantly, because surround suppression is present in parasol retinal ganglion cells[13,31], it is clear that LGN magnocellular neurons inherit a portion of their surround suppression from retinal inputs. Local thalamic circuitry[32] and CG feedback[33,34] may also contribute to extraclassical surround suppression. There is still debate regarding relative contributions of feedforward (e.g. retinal), local circuit, and feedback inputs to surround suppression at the level of the LGN and within the cortex. Notably, a recent study employing causal methods to suppress corticocortical feedback showed that inactivating cortical feedback to V1 reduced surround suppression and increased receptive field sizes[35]. Thus, in addition to identifying potential stream-specific functions for CG feedback, we also explored whether CG feedback makes a distinct functional contribution to extraclassical surround suppression in the LGN.

To assess the functional contribution of CG feedback in extraclassical surround suppression and thereby test whether CG feedback is functionally stream-specific or homogenous, we selectively manipulated CG circuits in anesthetized macaque monkeys via optogenetics while recording the responses of LGN neurons to a comprehensive battery of visual stimuli designed to evoke distinguishable responses across neuronal types. We demonstrate that CG feedback finely tunes extraclassical surround suppression for magnocellular neurons (but not parvocellular neurons), sharpening their receptive fields and generating an increase in their preferred spatial frequency. In a cross-species comparison, we find similar effects among Y LGN neurons (but not X neurons) in ferrets. Thus, in highly visual mammals, CG feedback exerts stream-specific influence over spatial resolution in the magnocellular/Y streams. Importantly, CG feedback also improves the temporal precision of LGN responses generally. Together these results suggest dual roles for CG circuits in both spatial and temporal precision, with spatial refinement occurring in a stream-specific manner. Dual functions for corticothalamic feedback reconcile prior findings across species and sensory modalities and suggest that these critical sensory feedback circuits are more functionally sophisticated than previously thought.

## Results

To examine the functional contributions of CG feedback to LGN visual responses, we utilized virus-mediated gene delivery to selectively express a light-sensitive ion channel (ChR2) or chloride pump (ArchT) within CG neurons following injection of a retrograde-only modified rabies virus into the LGN (Fig. 1a). In three monkeys, virus injections were within caudal portions of the LGN, corresponding to parafoveal eccentricities (Fig. 1b) and virus-labeled CG neurons were observed in V1 (Fig. 1c). All virus-labeled CG neurons had cell bodies located within layer 6, in both upper and lower tiers of layer 6, consistent with prior

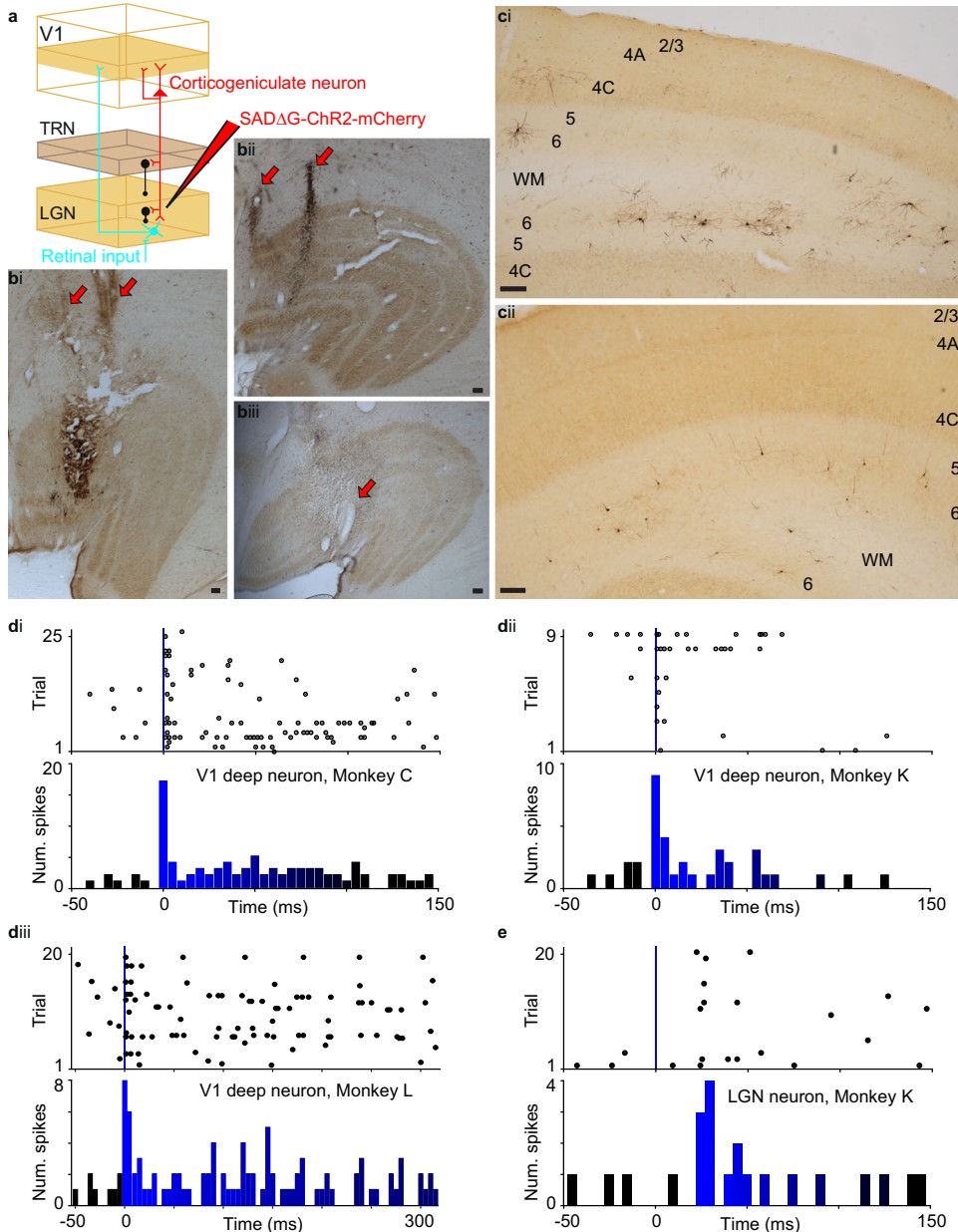

**Fig. 1 | Virus infection and LED-only stimulation of CG neurons. a** Schematic representation of virus injection into LGN and retrograde infection of CG neurons (red) in V1. Retinal inputs to LGN relay neurons (both cyan) and inhibitory LGN and TRN inputs (black) are also shown. **b** Coronal sections through caudal portions of the LGN in Monkeys L (i), C (ii), and K (iii) stained for cytochrome oxidase to reveal LGN layers and against mCherry to reveal virus injection sites. Red arrows mark injection needle penetration tracks. Medial is left, dorsal is up in each. Scale bars are 200 microns. **c** Coronal sections through V1 in Monkeys L (i) and K (ii) stained for cytochrome oxidase (layers labeled) and against mCherry to reveal virus-infected CG neurons. Scale bars are 200 microns. **d** Rasters and PSTHs of putative CG neurons recorded in Monkeys C (i), K (ii), and L (iii) in response to LED stimulation alone. Blue lines in rasters illustrate LED onset. LED protocol involved multiple pulses (including after time 0) in diii prompting the longer x-axis scale. **e** Raster and PSTH of an LGN neuronal response to LED stimulation alone; conventions as in (**d**).

findings[4,36–38], and demonstrated morphological characteristics of CG neurons in macaque V1[39]. Between 31 and 46% of virus-labeled CG neurons per monkey had cell bodies positioned in lower layer 6, indicating that virus injections in all three monkeys targeted both parvocellular and magnocellular layers of the LGN. No labeled axons were present in layer 4C (Fig. 1c), confirming that LGN relay cells were not infected by the virus.

## LED-driven responses in V1 and LGN

About 1 week following surgical injection of virus into the LGN, animals were anesthetized and paralyzed to prevent eye movements, and

recordings were made from LGN and opercular V1 neurons with overlapping receptive fields (Supplementary Fig. 1) using multi-electrode arrays. We first aimed to demonstrate electrophysiologically that putative CG neurons in V1 expressed ChR2 by evoking spiking responses during LED stimulation alone, in the absence of any visual stimulation. We identified 2-7 V1 neurons per monkey that were driven by LED stimulation alone (Fig. 1d). All V1 neurons for which we observed short-latency, reliable spiking responses to LED stimulation alone were located deep in the cortex (average distance from pial surface = 1.9 ± 0.1 mm), suggesting these putative CG neurons were in layer 6. Hit rates for observing V1 neurons directly modulated by LED

stimulation alone – relative to the "best-case" likelihood of encountering ChR2-expressing CG neurons given CG density in layer 6, number of recording penetrations per monkey, and electrode array configuration – were 1-to-3.5 times the best-case sample rate across monkeys. Interestingly, 8 neurons directly modulated by LED stimulation alone were also observed in the LGN across monkeys (see Fig. 1e for an example). LED-only activation of LGN neurons was transsynaptic via CG inputs because the action of rabies virus is exclusively retrograde, i.e. LGN relay neurons were not driven by antidromic LED activation of their axons in V1 as these are not labeled by the virus (Fig. 1c). Several observations support this idea. First, latencies between LED onset and LGN neuronal spiking responses were between 5 and 15 msec, reflecting the axon conduction times of stimulated CG neurons converging upon LGN neurons[40]. Second, receptive field center separations for simultaneous LGN and V1 recordings in which LED-only modulated LGN and/or V1 neurons were observed were slightly, albeit significantly, smaller than receptive field separations for other recordings in which LED-only modulation was not observed (p = 0.029, rank sum test; separation for pairs with LED-onlymodulated neurons = 0.17 ± 0.11 degrees, for other pairs=0.67 ± 0.15 degrees; Supplementary Fig. 1). Third, LED-mediated increases in stimulus-evoked firing rate across LED-only modulated LGN neurons were similar across monkeys (average firing rate fold change for 4 LED-only

modulated LGN neurons in Monkey K = 1.16 ± 0.15; for 3 LED-only modulated LGN neurons in Monkey C = 1.18 ± 0.26; and for 1 LED-only modulated LGN neuron in Monkey L = 1.14 ± 0.06). Together these findings suggest that LED stimulation of the V1 cortical surface activated ChR2 in virus-infected CG neurons with sufficient efficacy to drive spiking responses in these CG neurons as well as in some LGN neurons receiving convergent input from CG neurons with highly overlapping receptive fields. Furthermore, distributions of virus-labeled CG cell bodies throughout layer 6 and LED-evoked responses across LGN neurons per monkey suggest similar virus expression and optogenetic activation patterns across animals.

### Optogenetic effects on LGN neuronal tuning

Having demonstrated LED stimulation of putative CG neurons in V1, we next tested whether optogenetic stimulation of CG feedback modulated visual responses of LGN neurons in a stream-specific or generalized manner. Probing stream-specificity required the use of a comprehensive battery of visual stimuli designed to elicit different responses across LGN neuronal types. We therefore measured LGN neuronal responses to drifting sinusoidal gratings varying in contrast, spatial frequency, temporal frequency, and size (Fig. 2), luminance and cone-modulating m-sequence stimuli, and flashing black spots, all under conditions without and with LED stimulation of CG feedback. A

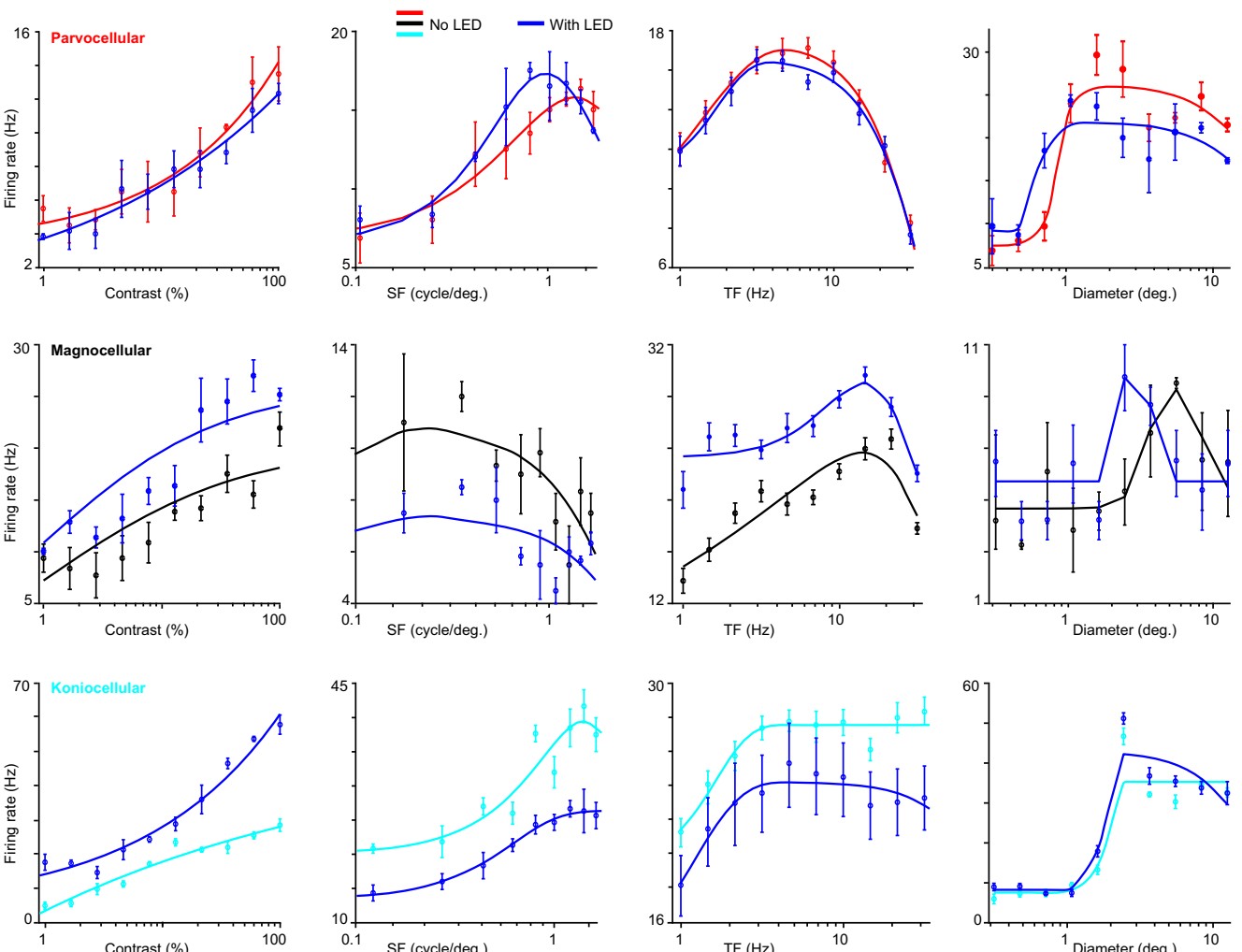

**Fig. 2 | Tuning curves for representative LGN neurons.** LGN neuronal responses to drifting gratings increasing in contrast (left column), spatial frequency (SF; second from left column), temporal frequency (TF; second from right column), and size/diameter (right column) in visual stimulation alone conditions (red – parvocellular neuron, top row; black – magnocellular neuron, middle row; cyan – koniocellular neuron, bottom row) and in visual stimulation synchronized with LED stimulation conditions (blue). Data are dots, error bars are SEMs, curves are based on fits (see Methods).

total of 98 LGN neurons across three monkeys met our criteria for visual responsivity (see Methods; 88 LGN neurons recorded with ChR2 stimulation and 10 LGN neurons recorded with ArchT stimulation). We characterized 33 parvocellular, 38 magnocellular, and 16 koniocellular LGN neurons as well as 11 LGN neurons of unknown type (only responses to flashing black spots were recorded). Classification of parvocellular, magnocellular, or koniocellular type was straightforward given responses across the comprehensive stimulus set. Accordingly, clustering and PCA analyses based on tuning metrics separated LGN neurons similarly to our subjective classification (Supplementary Fig. 2). Additionally, LGN neurons of the same type recorded in different monkeys showed remarkable similarity in their tuning properties measured across multiple stimulus dimensions (Supplementary Fig. 2), so data from all monkeys were combined for all analyses.

The first step was to ensure that our comprehensive stimulus set did indeed evoke distinguishable responses across LGN neurons of each type. As expected, parvocellular, magnocellular, and koniocellular LGN neurons demonstrated unique combinations of tuning responses to drifting gratings varying in contrast, spatial frequency (SF), temporal frequency (TF), and size (Fig. 2). We computed the following tuning metrics to quantify tuning differences across LGN neuronal types: c50 or the contrast to evoke a half-maximal response, preferred SF, preferred TF, TF high50 or the high TF to evoke a half-maximal response, preferred grating size (diameter), and SSI or the extraclassical surround suppression index. We also computed receptive field size (area) from luminance STAs (see Methods). As expected, magnocellular LGN neurons had significantly lower c50 and preferred SF values and significantly higher SSI values compared to parvocellular neurons (Fig. 3a, b, g, distributions above scatter plots; see Table 1 for all statistics, bottom row shows p values for comparisons across neuronal types). Additionally, magnocellular neurons had significantly higher preferred TF and TF high50 values relative to both parvocellular and koniocellular neurons (Fig. 3c, d, distributions above scatter plots; Table 1). We did not observe significant differences in spontaneous or visually evoked firing rates across neuronal types, although there was a trend whereby magnocellular neurons had higher spontaneous firing rates (Table 2).

To explore possible stream-specific impacts of CG feedback on LGN neurons, we next examined the effects of CG feedback on LGN tuning by comparing tuning metrics across conditions without and with LED stimulation of CG feedback (Fig. 3). Optogenetic stimulation of CG feedback did not alter c50, preferred TF, or TF high50 values for any LGN neuronal types (Fig. 3a, c, d; see Table 1 for all statistics, p values for comparisons across LED conditions labeled at left). Also, although individual neurons often showed increases or decreases in response amplitude with LED stimulation of CG feedback (Fig. 2, see magnocellular and koniocellular LGN neuron examples), these gain changes measured from neuronal tuning responses were inconsistent, even within the same neuron. To quantify gain changes across LED conditions, we computed the gain for SF and TF tuning curves as the area under the curve. There were no significant optogenetic effects on gain for any LGN neuronal type (Supplementary Fig. 3, Table 2).

Instead, significant effects of LED stimulation of CG feedback were observed for several tuning metrics for magnocellular LGN neurons only: preferred SF values were significantly increased (Fig. 3b, distribution right of scatter plot; Table 1), preferred grating size as well as receptive field size were significantly reduced (Fig. 3e, f, distributions right of each scatter plot; Table 1), and SSI was significantly increased (Fig. 3g, distribution right of plot; Table 1). Importantly, LED effects on SSI for magnocellular neurons were significant in each individual monkey (p < 0.05 for all). Similarly, optogenetic effects on stimulus size preferences were consistent across neurons in the sample, i.e. most magnocellular neurons shifted their preferences toward higher SFs (Fig. 4a), smaller grating sizes (Fig. 4b) and receptive field sizes

(Fig. 4c), and higher SSI values (Fig. 4d) with LED stimulation of feedback. Moreover, these effects are probably linked: by increasing extraclassical surround suppression, the classical receptive field area, and thus preferred stimulus size, shrink leading to an increase in the preferred grating spatial frequency (Fig. 5). Therefore, CG feedback is functionally stream-specific in the spatial domain, regulating receptive field size through extraclassical suppression for the magnocellular stream selectively.

## Optogenetic suppression of CG feedback

Given that ChR2-mediated activation of CG feedback increased extraclassical surround suppression, shrunk receptive field sizes, and increased preferred grating spatial frequencies among magnocellular, but not parvocellular or koniocellular LGN neurons, we next asked whether ArchT-mediated inhibition of CG feedback might have opposing effects in magnocellular LGN neurons. We recorded from 10 LGN neurons in one hemisphere of Monkey L in which a virus containing ArchT was injected into the LGN to infect V1 CG neurons. Although our sample size was small, we observed a significant reduction in SSI among magnocellular LGN neurons, but no change in preferred SF or grating size (Supplementary Fig. 4). Our observations of bidirectional effects of optogenetic stimulation/inhibition on SSIs suggest that CG feedback plays a key role in fine-tuning extraclassical surround suppression in a stream-specific manner.

## Optogenetic effects on LGN neuronal tuning in ferrets

Magnocellular stream-specific effects of CG feedback on spatial receptive field properties in monkeys prompted us to test whether similar effects are present across species, especially in highly visual carnivores. We examined X and Y LGN neurons recorded across 5 ferrets in which virus injections into the LGN resulted in ChR2 expression among CG neurons in area 17. The same tuning metrics based on neuronal responses to drifting gratings were used to classify X and Y neurons according to their pattern of tuning responses (see Methods). Similar to monkey magnocellular LGN neurons, ferret Y neurons preferred significantly higher SFs and had greater SSI values with LED stimulation of CG feedback (Supplementary Fig. 5; see Supplementary Table 1 for all statistics). As in monkeys, these optogenetic effects were not observed among ferret X neurons. Interestingly, there was a non-significant trend whereby optogenetic stimulation of CG feedback increased TF high50 values among Y neurons (Supplementary Fig. 5; Supplementary Table 1), an effect similar to one reported previously[22]. In summary, across two species (macaques and ferrets), CG feedback exerts stream-specific effects on the spatial receptive fields of LGN neurons (magnocellular and Y) that demonstrate extraclassical surround suppression.

## Optogenetic effects on spatiotemporal receptive fields

In addition to modulations of spatial resolution during corticocortical feedback manipulation, a prior study also reported changes in receptive field response amplitude[35]. Although we did not observe optogenetic effects on amplitude measured from tuning curves, we asked whether LED activation of CG feedback altered amplitudes for LGN responses to white noise stimuli (Fig. 6). Consistent with the prior study, magnocellular neuronal receptive field amplitudes from white noise STAs were significantly larger with LED stimulation of CG feedback (p = 0.0027, paired t-test; average amplitude without LED stimulation = 0.37 ± 0.9 [arb. units], average amplitude with LED = 0.44 ± 0.9 [arb. units]). Thus, part of the stream-specific influence of CG feedback on spatial resolution includes widening the dynamic range over which magnocellular neurons detect luminance contrast.

White noise STAs also provided additional insights into CG influence over LGN responses in the temporal domain. Response latency from temporal STAs was computed as the time corresponding to the

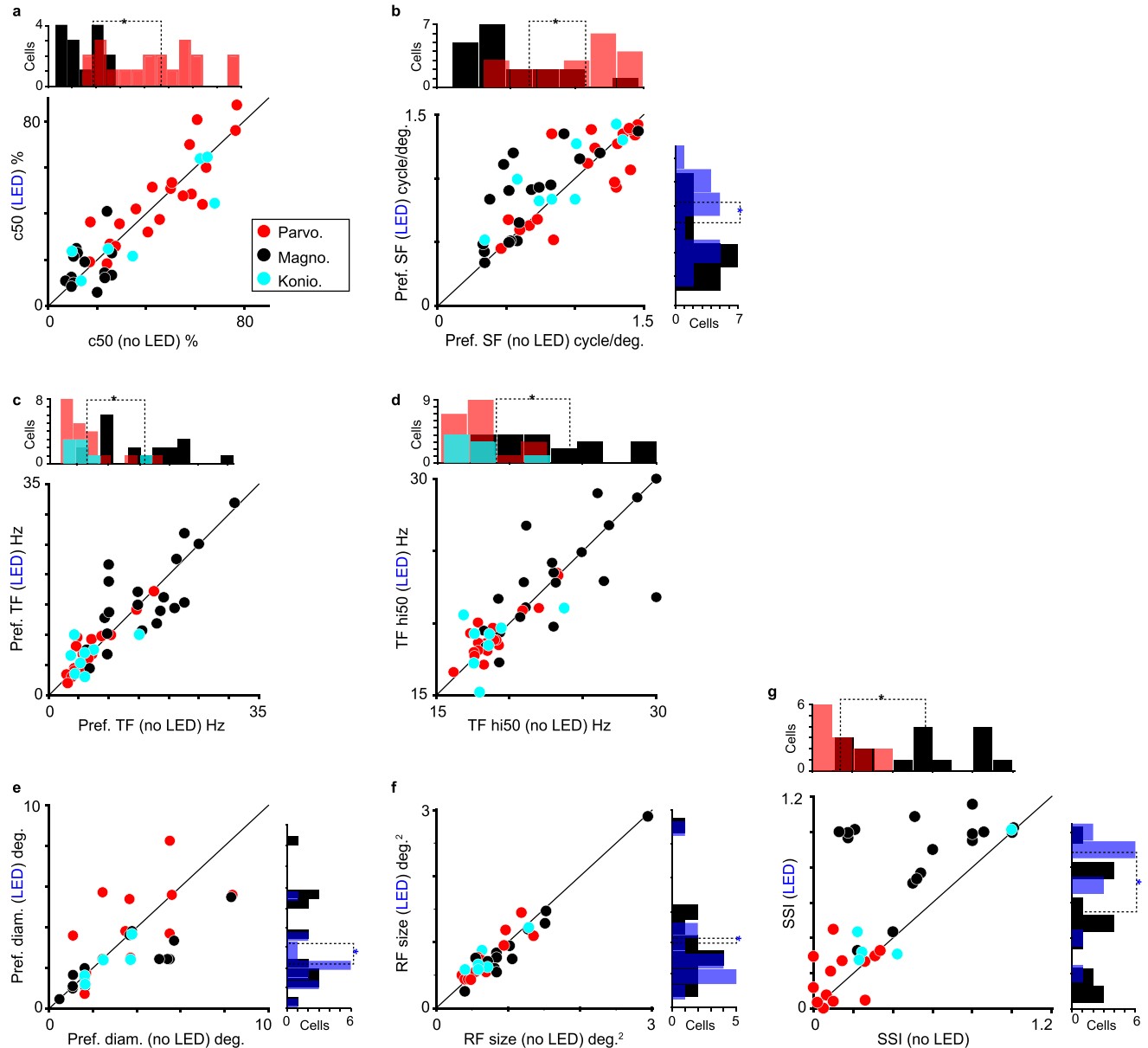

**Fig. 3 | Tuning metrics differ between LGN neuronal types and across LED conditions.** Tuning metrics for all sampled parvocellular (red), magnocellular (black), and koniocellular (cyan) LGN neurons across conditions without and with LED stimulation of CG feedback: c50 (**a**), preferred spatial frequency (SF, **b**), preferred temporal frequency (TF, **c**), TF high50 (TF hi50, **d**), grating diameter to evoke maximum response (**e**), receptive field (RF) size measured from achromatic m-sequence stimuli (**f**), and surround suppression index (SSI, **g**). Distributions above plots illustrate differences between all sampled parvocellular (and koniocellular in **c**, **d**) and all magnocellular neurons, dashed lines indicate averages per cell type, black asterisks indicate significant differences (see Table 1 for statistics). Distributions to the right of plots illustrate differences across LED conditions for sampled magnocellular neurons only, dashed lines indicate averages per condition, blue asterisks indicate significant differences (see Table 1 for statistics). Source data are provided in Source Data file.

brightest STA pixel relative to the time of the spike (time = 0 in Fig. 6a–c, right curves). Fig. 6d illustrates these latencies measured per LGN neuron from STAs without and with LED stimulation of CG feedback (also compare top to bottom row STAs and black/red/green/cyan vs blue pixel brightness curves in Fig. 6a–c for example LGN neurons of each type). Across all LGN neurons with m-sequence data, response latency was significantly reduced when CG feedback was optogenetically stimulated (p = 0.043, paired t-test; average latency no LED = 29.32 ± 2.03 msec, average latency with LED = 27.47 ± 1.57 msec, n = 33 neurons; Fig. 6d). In contrast to stream-specific effects on spatial resolution and STA amplitude, a generalized effect on response latency suggests that CG feedback regulates the timing of LGN responses more generally, i.e., across cell types.

## Optogenetic effects on LGN neuronal response precision

We therefore tested whether LED stimulation of CG feedback alters the temporal precision of LGN neurons using a different dynamic stimulus: a flashing spot. Prior studies have shown that CG feedback regulates the temporal precision and visual response variability of LGN neurons, with similar effects noted across LGN neuronal types[22,24,25,41]. Accordingly, we specifically examined whether CG feedback modulates LGN temporal precision in a generalized or stream-specific manner. Flashing black spots produced short-latency responses to spot onset among LGN Off neurons and longer latency responses (to spot offset) among On neurons (Fig. 7a, b). We did not observe differences in response latencies to flashing spots across LGN neuronal types (magnocellular, parvocellular, koniocellular, On or Off) or across conditions without or

**Table 1 | Tuning metrics for LGN neurons**

| | c50 (%) No LED / with LED | Preferred SF (cycles/deg.) | Preferred TF (Hz) | TF high50 (Hz) | Preferred size, grating (deg.) | SSI | Receptive field size, m-seq. (deg².) |
|---|---|---|---|---|---|---|---|
| Parvo neurons (n = 20) | 45.7 ± 4   43.3 ± 5 | 1.1 ± 0.1   1.1 ± 0.1 | 6.9 ± 0.8   7.3 ± 0.8 | 19 ± 0.4   19.2 ± 0.4 | 3.5 ± 0.6   3.8 ± 0.7 | 0.14 ± 0.04   0.19 ± 0.04 | 0.7 ± 0.07   0.7 ± 0.08 |
| LED effect: P-values (paired t-test) | 0.59 | 0.92 (n = 13) | 0.19 | 0.27 | 0.62 (n = 13) | 0.22 (n = 13) | 0.56 (n = 17*) |
| Magno neurons (n = 20) | 20.2 ± 3   20.9 ± 4 | 0.65 ± 0.07   0.82 ± 0.08 | 15.8 ± 2   15.9 ± 2 | 23.6 ± 1   23.1 ± 1 | 3.3 ± 0.6   2.3 ± 0.3 | 0.55 ± 0.07   0.88 ± 0.06 | 1.1 ± 0.18   1 ± 0.18 |
| LED effect: P-values (paired t-test) | 0.37 (n = 17) | 0.0022 (n = 18) | 0.94 | 0.55 | 0.0109 (n = 17) | $3.5 \times 10^{-4}$ (n = 17) p < 0.01 Wilcoxon | 0.0378 (n = 14*) |
| Konio neurons (n = 8) | 35 ± 10   32 ± 9 | 0.9 ± 0.1   1 ± 0.1 | 6.5 ± 1.4   6.7 ± 1 | 18.8 ± 1   18.8 ± 1 | 2.7 ± 0.5   2.3 ± 0.5 | 0.42 ± 0.2   0.47 ± 0.2 | 0.7 ± 0.1   0.8 ± 0.1 |
| LED effect: P-values (paired t-test) | 0.48 | 0.14 | 0.92 | 0.98 | 0.19 (n = 5) | 0.4 (n = 5) | 0.37 (n = 6*) |
| Neuronal type differences (non-parametric ANOVA) | 0.0003 Parvo > Magno | 0.0009 Parvo > Magno | $8.4 \times 10^{-6}$ Magno > Parvo & Konio | $4.8 \times 10^{-5}$ Magno > Parvo & Konio | 0.5 | 0.0028 Magno > Parvo | 0.0749 |

Average contrast to evoke a half-maximal response (c50), preferred spatial frequency (SF), preferred temporal frequency (TF), temporal frequency to evoke a half-maximal response greater than the preferred frequency (TF high50), preferred grating size, surround suppression index (SSI), and receptive field size for parvocellular (parvo), magnocellular (magno), and koniocellular (konio) LGN neurons; for each metric, average values without LED at left, with LED at right (LED labels only shown for c50 columns). Some tuning metric averages were computed from a subset of neurons (n's indicated below p values). Asterisks indicate different subsets of neurons contributing to receptive field size measurements from white noise m-sequence data compared to neurons contributing to grating-based tuning metrics. P-values indicate significant effects of the LED (paired t-tests). Wilcoxon signed-rank test noted), using a Bonferroni corrected alpha=0.0125 for contrast, SF and TF metrics and a Bonferroni corrected alpha=0.025 for size metrics. Bottom p-values indicate significant differences across neuronal types (comparisons of no LED responses).

with LED stimulation of CG feedback (Table 3), likely because LGN neurons responded reliably to this stimulus. In contrast, response reliability, or the probability that a neuron spiked at the same latency across trials, increased significantly with LED stimulation of CG feedback (Fig. 7c, see Table 3 for statistics). Significant increases in reliability were present for On and Off LGN neurons, although the effect was larger for On neurons. Reliability was also significantly increased with feedback activation for parvocellular neurons alone (Table 3). Consistent with an increase in reliability, LED stimulation of CG feedback also significantly reduced jitter, a measure of response variability, among all LGN neurons (Fig. 7d). Both On and Off LGN neurons showed significantly reduced jitter, although the effect was again larger for On neurons. Jitter was also significantly reduced for magnocellular neurons alone (Table 3). LED effects on reliability and jitter were also significant in two monkeys individually (p < 0.05 for all).

In addition to measures of reliability and jitter, which quantify the first stimulus-driven spiking responses of neurons, we also asked whether spiking patterns measured over a wider response window (10 msec) were altered by LED stimulation of CG feedback. We observed a significant increase in the probability of patterns including a single spike with LED stimulation of CG feedback (Fig. 7e), akin to an increase in spike occurrence following the stimulus. This effect was significant across all LGN neurons but was driven by On neurons (Table 3). Together these results are consistent with prior findings and indicate that CG feedback improves the temporal precision of LGN neuronal responses by reducing spiking variability. Importantly, effects of CG feedback on temporal precision were generalized across LGN neuronal types.

As a final confirmation of stream-specific effects of CG feedback, we examined whether optogenetic effects were correlated. We hypothesized that since effects of LED activation of CG feedback on spatial resolution were magnocellular-stream specific while effects of CG feedback on temporal precision were generalized, LED-induced changes in spatial tuning metrics would not be correlated with LED-induced changes in temporal metrics across LGN neurons. We found no significant correlations between LED modulations of neuronal tuning and response reliability, nor did we observe correlated LED effects across neuronal tuning metrics. These results further indicate that the significant spatial resolution effects we observed were indeed magnocellular-stream specific, while effects on temporal precision were generalized.

As described above, LED activation of CG feedback increased the amplitude of magnocellular neuronal responses in white noise STAs, but there were no significant LED effects on neuronal firing rate or response gain measured from tuning curves (Table 2, Supplementary Fig. 3). To address the possibility that LED activation of CG feedback altered rhythmic activity in CG circuits, we analyzed power spectra of LGN and V1 neuronal spiking and local field potentials in response to LED stimulation alone (in the absence of visual stimulation) and comparing visual stimulation alone to visual stimulation synchronized with LED stimulation. LED stimulation alone produced peaks in power spectra at frequencies corresponding to LED flash frequencies, but there were no differences across power spectra for visual stimulation alone trials compared to visual stimulation synchronized with LED stimulation trials. These findings suggest that some V1 and LGN neurons were directly LED-modulated due to expression of ChR2, but that visual stimulation was stronger than optogenetic activation alone. In support of this idea, there was a significant correlation between LED effects on maximum evoked firing rates and spontaneous firing rates across LGN neurons ($R^2 = 0.4$, $p = 8.6 \times 10^{-7}$, linear regression). Additionally, there was a weak but significant positive correlation between LED effects on maximum evoked firing rates and overall response gain measured from temporal frequency tuning curves across LGN neurons ($R^2 = 0.26$, $p = 0.002$, linear regression). In other words, a subset of LGN neurons that were more strongly LED-modulated showed

**Table 2 | Firing rates and gains for LGN neurons**

| | Spontaneous FR (Hz) No LED / with LED | | Max. evoked FR (Hz) | | AUC for SF curves (arb. units) | | AUC for TF curves (arb. units) | |
|---|---|---|---|---|---|---|---|---|
| Parvo neurons (n = 20) | 6.8 ± 1.3 | 6.4 ± 1.1 | 11.2 ± 2 | 11.6 ± 2 | 63 ± 12 | 64 ± 13 | 118 ± 23 | 147 ± 34 |
| LED effect: P-values (paired t-test) | 0.27 | | 0.73 | | 0.82 | | 0.21 | |
| Magno neurons (n = 20) | 12 ± 2.6 | 12.9 ± 3.2 | 14.3 ± 2.6 | 15.7 ± 3.2 | 65 ± 14 | 56 ± 11 | 73 ± 14 | 70 ± 18 |
| LED effect: P-values (paired t-test) | 0.3 | | 0.19 | | 0.29 | | 0.79 | |
| Konio neurons (n = 8) | 7.1 ± 1.6 | 7.3 ± 1.9 | 14.3 ± 3.3 | 12.8 ± 2.8 | 89 ± 26 | 78 ± 18 | 124 ± 34 | 124 ± 29 |
| LED effect: P-values (paired t-test) | 0.7 | | 0.22 | | 0.92 | | 0.99 | |
| Neuronal type differences (non-parametric ANOVA) | 0.09 | | 0.5 | | 0.5 | | 0.2 | |

Average spontaneous firing rate (FR), maximum visually evoked firing rate, area under the curve (AUC) for spatial frequency (SF) and temporal frequency (TF) tuning curves for parvocellular (parvo), magnocellular (magno), and koniocellular (konio) LGN neurons; for each metric, average values without LED at left, with LED at right (LED labels only shown for spontaneous FR columns). No significant differences observed across LED conditions (paired t-tests) or across neuronal types (comparisons for no LED responses).

consistent LED-mediated effects on firing rate measures and response gain. Also, the handful of LGN neurons that were directly modulated by LED stimulation of CG feedback showed slightly larger fold-changes in STA response amplitude and maximum evoked and spontaneous firing rates relative to the LGN population (average fold-change in: amplitude = 1.64 ± 0.44 [n = 5 LED-modulated] vs. 1.03 ± 0.05 [n = 32]; max firing rate=1.08 ± 0.1 [n = 6 LED-modulated] vs. 1.06 ± 0.05 [n = 46]; spontaneous firing rate=1.06 ± 0.11 [n = 6 LED-modulated] vs. 1.01 ± 0.04 [n = 46]). Taken together, these findings suggest that LED stimulation of CG feedback was localized such that LGN neurons with more convergent input from ChR2-expressing CG neurons demonstrated consistently larger firing rate gain modulations. Thus, although LED activation of CG feedback did not produce significant shifts in firing rate or response gain across the full sample, a subpopulation of LGN neurons, presumably those receiving convergent input from ChR2-expressing CG neurons, showed consistent LED facilitation.

## Discussion

There have been numerous studies of corticogeniculate (CG) influence over LGN responses, many of which employed causal manipulations to explore the contributions of CG feedback to visual perception, reviewed in ref. 2. Evidence supporting stream-specific effects of CG feedback has been limited. Similarly, it was not clear whether CG feedback contributed to surround suppression in the LGN. This is surprising because in highly visual mammals such as primates and carnivores, the defining organizational principle of early visual circuits, both feedforward and feedback, is their parallel organization into functionally distinct processing streams. Why would these early visual circuits remain anatomically segregated and physiologically distinct if the first feedback step in the processing hierarchy was functionally stream-agnostic? In this study we demonstrate clear stream-specific effects of CG feedback in both primates and carnivores. Magnocellular/Y CG circuits play a "dual" role in regulating the feedforward flow of visual signals through the LGN: they sharpen the spatial receptive fields of their recipient LGN targets through enhancement of extra-classical surround suppression and they improve the temporal precision of LGN neuronal responses. Other CG circuits (parvocellular/X and koniocellular) also improve the temporal precision of their target LGN neurons. Accordingly, all CG circuits maintain control in the temporal domain, while magnocellular/Y CG circuits also regulate LGN neurons in the spatial domain.

Why have stream-specific effects of CG feedback been so difficult to uncover until now? There are likely many explanations. One principal explanation may be technical limitations. Until the advent of optogenetics and virus-mediated selective circuit tracing methods, it was not possible to selectively target CG neurons for causal manipulation in highly visual mammals. Prior efforts involved global

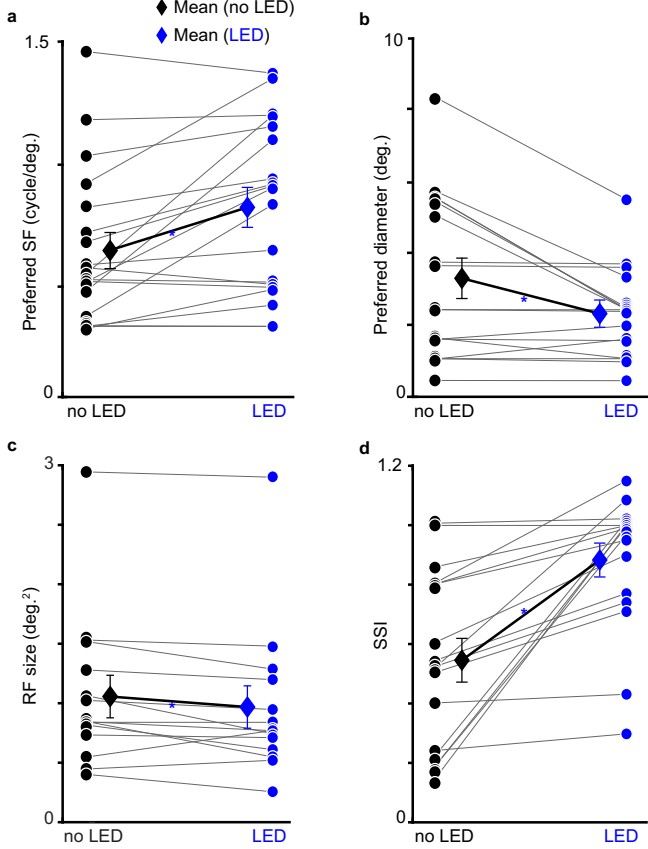

**Fig. 4 | Optogenetic effects on magnocellular neurons.** LED stimulation of CG feedback increased preferred spatial frequencies (SFs, **a**), reduced preferred grating size in diameter (**b**), also reduced receptive field (RF) size measured from achromatic m-sequence stimuli (**c**), and increased surround suppression index values (SSIs, **d**) consistently across magnocellular neurons in the sample. Gray lines link values without (black circles) and with (blue circles) LED stimulation per magnocellular neuron. Black/blue diamonds in the interior of each plot indicate magnocellular neuronal averages, error bars are SEMs, and blue asterisks indicate significant effects across LED conditions for magnocellular neurons (see Table 1 for statistics).

suppression of V1 through cortical cooling, lesioning, or pharmacological agent infusion e.g[24,25,42]. These methods suppress all CG circuits and all other circuits involving V1 neurons, which could have disproportional effects on different CG neuronal types. Importantly, our optogenetic manipulation of CG circuits was spatially targeted, as LED

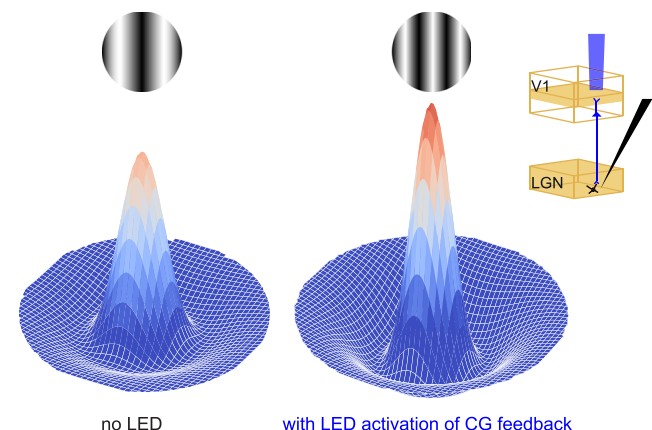

no LED          with LED activation of CG feedback

**Fig. 5 | Schematic representation of coordinated effects of optogenetic activation of CG feedback.** Without LED activation of CG feedback, magnocellular LGN neurons exhibit extraclassical surround suppression (dark blue depression) and prefer slightly lower spatial frequency gratings; with LED activation of CG feedback (inset top right), recorded magnocellular neurons exhibit increased surround suppression (deeper dark blue depression), have smaller classical receptive fields with larger response amplitude (narrower and taller central peak), and thus prefer higher spatial frequency gratings.

effects on firing rate and tuning gain were correlated. We attribute these correlated effects to the spatial overlap between LED-activated CG neurons and their converging target LGN neurons[5], as analogous effects have been observed in mouse LGN during corticothalamic optogenetic manipulation[27,43]. Advantageously, we conducted both optogenetic activation and suppression of CG circuits, an approach that is standard in mice e.g[27]., and observed bidirectional effects on magnocellular neuronal spatial precision, increasing the robustness of our findings. Of course, optogenetic circuit manipulation is sub-optimal: it is not natural, it may evoke poly-synaptic responses, and it does not allow differential activation of distinct CG neuronal types/circuits. Poly-synaptic LED-driven effects in LGN were unlikely to confound our findings of stream-specific effects on magnocellular neurons because local circuits involving magnocellular-, parvocellular-, and koniocellular-targeting CG neurons are distinct[44]. Results of optogenetics experiments could also be confounded by differential expression of opsins across subjects. Here, we demonstrate consistent expression and activation patterns across animals. Accordingly, our observation that parvocellular LGN neurons did not show any optogenetic enhancement of receptive field spatial resolution cannot be attributed to lack of optogenetic expression. Lastly, unnatural LED stimulation of CG neurons that may otherwise not spike synchronously could limit our ability to detect stream-specific effects of CG feedback. I.e., it might be possible to reveal even more stream-specific properties of CG feedback if we could deploy CG-circuit-specific optogenetics approaches. Our observation of robust stream-specific effects of CG feedback on LGN spatial resolution, despite these limitations, is encouraging for future efforts.

Another possible explanation for minimal stream-specific effects of CG feedback observed previously is the limited number of visual stimuli tested. Temporal effects of CG feedback are best revealed by stimuli with rapid onset/offset (like flashing, uniform-luminance spots), while extraclassical surround modulation requires gratings systematically varying in size. Most prior studies utilized smaller subsets of stimuli, precluding measures of temporal and spatial effects within the same experiment. Even wider ranges of stimuli may be necessary to resolve stream-specific effects of CG feedback. For example, two earlier studies that employed more naturalistic stimuli (including natural movies) in alert animals while inactivating V1

produced complex effects across LGN neurons, including reduced information content and increased sparseness in LGN spiking[25,28].

A further possible explanation for the minimal stream-specific effects observed previously could relate to species differences. The X/Y/W streams in carnivores are somewhat homologous to the parvocellular/magnocellular/koniocellular streams of primates, but this homology is not strict and there are notable differences, including cone-opponency, receptive field size ranges, transient versus sustained-ness of responses, and linear summation within LGN neuronal receptive fields, to name a few[45]. It is, therefore, noteworthy that we observed similar stream-specific effects of CG feedback on magnocellular/Y neurons across primates and carnivores, despite imperfect homology between these species. Interestingly, other effects in our study were not consistent across these two species. We and others had previously observed modulations of the gain of carnivore Y-neuronal responses to moving stimuli when CG feedback was manipulated[19,22]. Here, we report an analogous increase in the TF high50 for ferret Y neurons with optogenetic activation of CG feedback. However, we did not observe gain changes or increases in TF high50 for primate magnocellular neurons, although we did observe increases in STA amplitude for magnocellular neurons. In other words, it appears that Y-stream CG feedback in carnivores increases LGN Y neuronal response gain and preferences for faster-moving or higher temporal frequency stimuli, while magnocellular CG feedback in primates does not alter LGN neuronal response gain or preferences for stimulus motion. Why CG feedback should heighten LGN responses to stimulus motion in carnivores but not primates is an evolutionary question far beyond the scope of this study. But it nonetheless highlights the intriguing possibility that we are just scratching the surface of stream- and species-specific functions of CG circuits in visual perception. Increased understanding of stream- and species-specific functions of CG circuits may enable parallels to be drawn between other species, such as primates and rodents. Rodent retino-geniculo-cortical circuits contain multiple neuronal types that are rare and/or yet undiscovered in primates. So, while both species have parallel information processing streams, the qualities of visual information conveyed in those streams differ markedly. It is possible that more in-depth understanding of the stream-specific roles of CG circuits in primates (and carnivores) will shed light on the diverse effects of feedback manipulation observed to date in mice. Further stream-specific effects may be revealed by CG-circuit-selective manipulation and/or testing of additional visual stimuli that better isolate responses of different LGN neuronal types, and especially if these can be conducted in alert and behaving animals. Also, cross-species comparisons of CG circuit properties will produce important generalizable principles defining core functions of corticothalamic feedback across sensory modalities.

Our finding of stream-specific effects of CG feedback on magnocellular/Y LGN neuronal spatial resolution raises three key questions: 1) what is the circuit mechanism underlying this effect; 2) why does CG feedback sharpen magnocellular/Y neuronal spatial resolution; and 3) why does this function reside in the magnocellular/Y stream and not in the parvocellular/X stream? CG axons provide spatially-restricted synapses onto the distal dendrites of LGN relay neurons[5,46] and they also contact two types of inhibitory neurons: inhibitory neurons within the thalamic reticular nucleus (TRN) and local LGN inhibitory interneurons, both of which synapse onto LGN relay neurons[47]. Thus, CG axons can monosynaptically excite and disynaptically inhibit LGN relay neurons. While CG synapses onto LGN relay neurons are numerous relative to retinal inputs, they are small and weak, producing small EPSCs, and are therefore classified as modulating rather than driving synapses[9]. Given the modulatory nature of CG feedback, it follows that CG influence over LGN neuronal spatial resolution is also modulatory. Indeed, extraclassical surround suppression among LGN (mostly magnocellular) neurons is mediated largely by retinal inputs and local

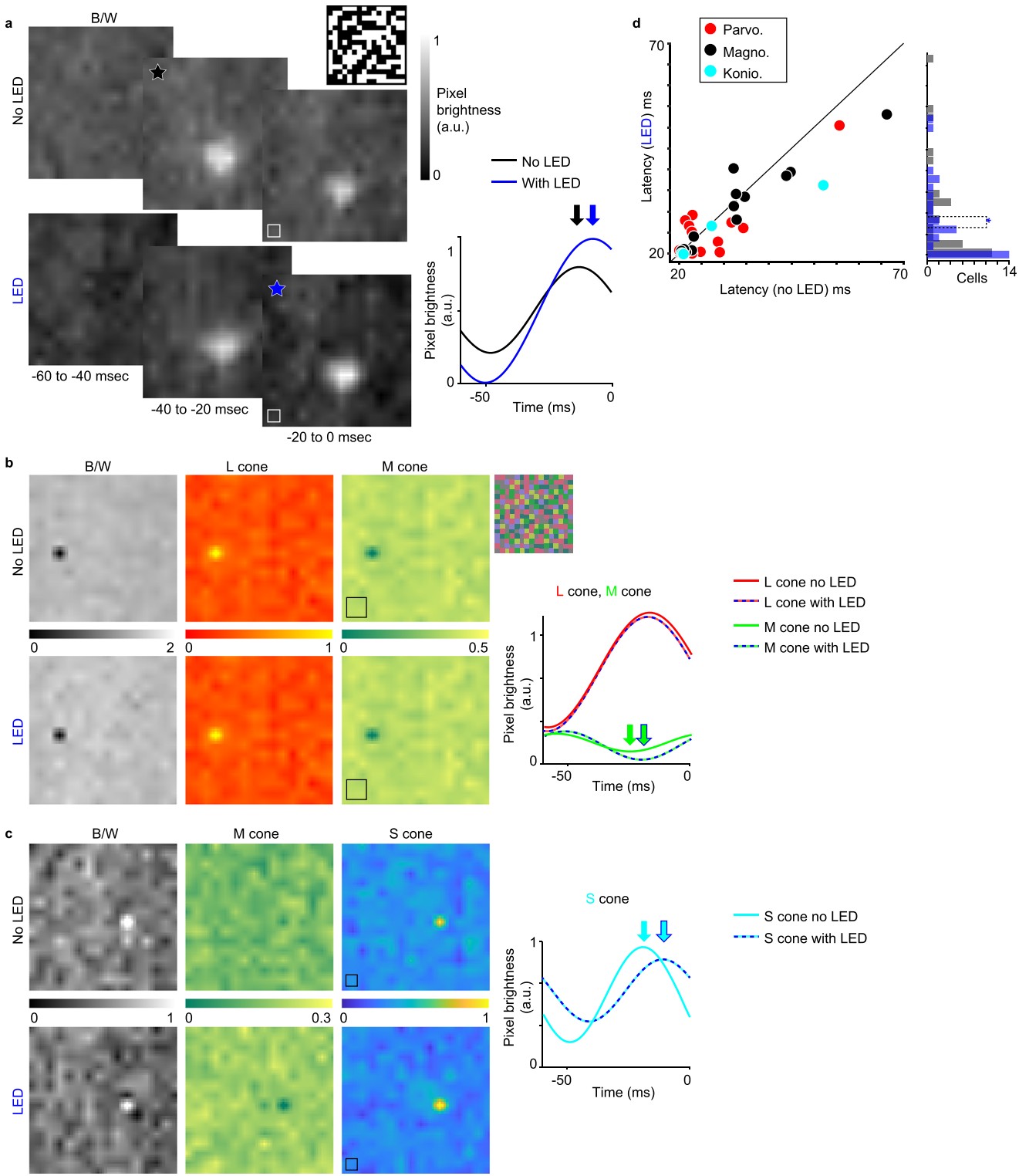

inhibition within the LGN that they recruit[13,32,48]. Accordingly, suppressing V1 (and CG feedback) increases the receptive field sizes of LGN neurons and reduces surround suppression, but does not eliminate it[27,49,50]. Importantly, the receptive fields of CG neurons are slightly larger than those of retinal and LGN neurons[33], providing a substrate for broader modulatory feedback. Additionally, CG influence over LGN neuronal responses is excitatory when their receptive fields are aligned and net inhibitory when they are misaligned[51,52]. Together, these findings suggest that broader modulatory CG signals with a

particular pattern of spatial alignment, mediated by monosynaptic excitatory input plus disynaptic inhibition, could underlie the effects of CG feedback on sharpening LGN spatial resolution.

Here we selectively manipulated CG neurons in V1. While CG neurons are also present in V2[39], these were not targeted by the LED, which was positioned many millimeters from the V1/V2 border. Given that corticocortical feedback to V1 regulates V1 neuronal spatial resolution[35] similarly to CG feedback to LGN, it is possible that cortical feedback plays a generalized, perhaps cascading role in improving

**Fig. 6 | Spatiotemporal receptive fields for representative LGN neurons. a** Three STAs around the peak frame (black/blue stars) for an On magnocellular neuron in response to the achromatic luminance (B/W; schematic of stimulus frame at top right) m-sequence stimulus alone (top row) and with LED stimulation (bottom row; STAs from both conditions scaled to same color bar). White boxes illustrate 0.5-degree squares. Curves at right illustrate temporal modulation in brightness for the maximum-value pixel within the receptive field without (black) and with (blue) LED stimulation. Arrows indicate peak responses and associated response latencies without (black) and with (blue) LED stimulation. **b** Peak frame STAs for an M-Off/L-On parvocellular neuron in response to the achromatic (B/W) stimulus as well as the cone-modulating stimulus (schematic of stimulus frame at top right, RGB values are approximations for display purposes) without (top) and with (bottom) LED stimulation. Black boxes illustrate 1-degree squares; STAs for both conditions scaled to same color bar in between. Right: temporal modulation in brightness for the maximum-value pixel (red) and minimum-value pixel (green) within the receptive field without (red/green) and with (dashed blue with red/green fill) LED stimulation.

Arrows indicate peak responses without (green) and with (blue/green) LED stimulation. **c** Peak frame STAs for an S-On/L&M-Off koniocellular neuron with conventions as in **b** except with S-cone activation illustrated. Curves at right show S-cone STA maximum-value pixel brightness temporal modulation without (cyan) and with (dashed blue with cyan fill) LED stimulation. Arrows indicate peak responses without (cyan) and with (blue/cyan) LED stimulation. **d** Scatterplot of peak response latencies (measured from peaks/troughs in the brightest pixel temporal modulation curves, as in **a** arrows) across LED conditions for all sampled parvo-cellular (red), magnocellular (black), and koniocellular (cyan) LGN neurons. Distribution at right of plot illustrates latencies for all sampled LGN neurons without (gray) and with (blue) LED stimulation of CG feedback, dashed line indicates average latencies per LED condition, blue asterisk indicates significant difference across LED conditions ($p = 0.043$, paired t-test (two-sided); average latency without LED = $29.3 \pm 2.03$ msec, with LED = $27.47 \pm 1.57$ msec). Source data are provided in Source Data file.

visual spatial resolution. However, we demonstrate clear stream-specific effects of CG feedback on the magnocellular stream. Therefore, any cascading effects of cortical feedback would need to be filtered through local V1 circuits. Additionally, it remains unknown whether corticocortical circuits include functionally heterogeneous neurons that may differentially regulate spatial and perhaps temporal resolution among their target neuronal populations.

Why might CG feedback need to sharpen LGN spatial resolution and why does this function reside in the magnocellular/Y stream? These questions and likely linked by the fact that neurons in the magnocellular stream tend to have larger receptive fields and demonstrate extraclassical surround suppression mediated through feedforward retinal circuits[31,53]. In contrast, parvocellular neurons have smaller receptive fields, especially in the fovea, to support high acuity vision, and show minimal extraclassical surround suppression[13,29,53]. It is possible that parvocellular neurons are already specialized for selectively processing small portions of the visual field because they are driven by inputs from single/few cones – i.e. they do not need the extra processing "layer" of extraclassical surround suppression. Since magnocellular neurons sum inputs from multiple cones, an extra layer of spatial sharpening may be required to ensure these neurons maintain some selectivity for stimulus size. Much of their surround suppression is inherited from the retina, but some comes from CG feedback, as discussed above. In further support of this notion, magnocellular LGN neurons show a larger change in response gain to small versus large sized stimuli when V1 is ablated relative to parvocellular neurons[34]. Also, surround suppression in magnocellular LGN neurons is contrast-dependent[54] just as magnocellular neurons are more sensitive to small stimulus changes at low contrast. Taken together, magnocellular CG feedback could provide a modulatory boost to feedforward extraclassical surround suppression that is needed more for magnocellular neurons because they sum inputs from multiple cones and are sensitive to low contrast stimuli. Additionally, the CG-mediated boost in spatial resolution complements feedforward surround suppression by providing signals from a larger portion of the visual field.

While CG-mediated improvement in spatial resolution may only be necessary for the magnocellular stream, the influence of CG feedback in improving the temporal precision of LGN responses appears to be necessary across streams. In other words, CG influence over LGN activity is more generalized and more robust in the temporal domain in the primate, as we have also shown previously in the carnivore[22]. The need to reduce variability and improve signal transmission fidelity in the face of internal neuronal noise is present throughout the sensory system. CG circuits likely accomplish jitter reduction and improved LGN response reliability through the combination of monosynaptic excitation and disynaptic inhibition. It is tempting to suggest that the two types of inhibitory microcircuits involved, through the TRN and

local LGN interneurons, may facilitate the dual mode of operation for CG circuits. TRN inhibition could, for example, be the basis for generalized temporal precision across CG circuits and LGN neuronal types. At the same time, local LGN inhibitory interneurons could support stream-specific CG feedback effects, like improving spatial resolution among magnocellular neurons. Testing these alternative hypotheses would require methods to selectively manipulate TRN or LGN interneurons in highly visual mammals.

Here we provide compelling evidence that CG circuits are functionally stream-specific, matching their clear anatomical and physiological segregation in highly visual mammals. Furthermore, we demonstrate that CG circuits operate in dual functional modes to improve both the temporal precision and spatial resolution of LGN responses to incoming visual information. This latter finding helps to reconcile inconsistent observations of corticothalamic function across sensory modalities. For example, corticothalamic circuits in the auditory and somatosensory systems have been shown to alter thalamic neuronal tuning in an "egocentric" manner[55,56]. An analogous effect in the visual system would involve shifting the spatial tuning of LGN receptive fields toward that preferred by CG feedback neurons. Here we demonstrate that CG circuits in the magnocellular stream sharpen the spatial resolution of their target LGN neurons, likely reflecting the strong extraclassical surround suppression that magnocellular-targeting CG neurons display[14]. In that sense, we demonstrate a similar type of "egocentric" selection in the primate and carnivore visual systems. As we discover increasing functional diversity among distinct corticothalamic circuits across species and sensory modalities, we may uncover similar defining principles guiding the functional contributions of these critical feedback circuits to sensory perception.

## Methods

Three adult macaque monkeys (*Macacca mulatta*) of both sexes (2 female, 1 male) aged 6-10 years were used in this study. All animal procedures were approved by the Institutional Animal Care and Use Committees at Dartmouth College and the University of Rochester and conformed to the guidelines set forth by the USDA and NIH. In order to express the optogenetic cation channel channelrhodopsin2 (ChR2) and the fluorescent marker mCherry selectively in corticogeniculate (CG) neurons in the visual cortex, a genetically modified rabies virus (SADΔG-ChR2-mCherry, titer range: $1.3 \times 10^8$ – $2.5 \times 10^9$, Salk Institute Viral Core, San Diego, CA) was injected into the dorsal lateral geniculate nucleus of the thalamus (LGN), where it was taken up by axon terminals at the injection site, including axons of CG neurons (Fig. 1a). G-deleted rabies virus acts like a retrograde tracer, traveling backwards along axons to infect cell bodies, because the glycoprotein (G) essential for trans-synaptic mobility in wild rabies virus is not endogenously expressed[57–59]. The virus replicates within infected neurons resulting in robust expression of ChR2 and mCherry translated from

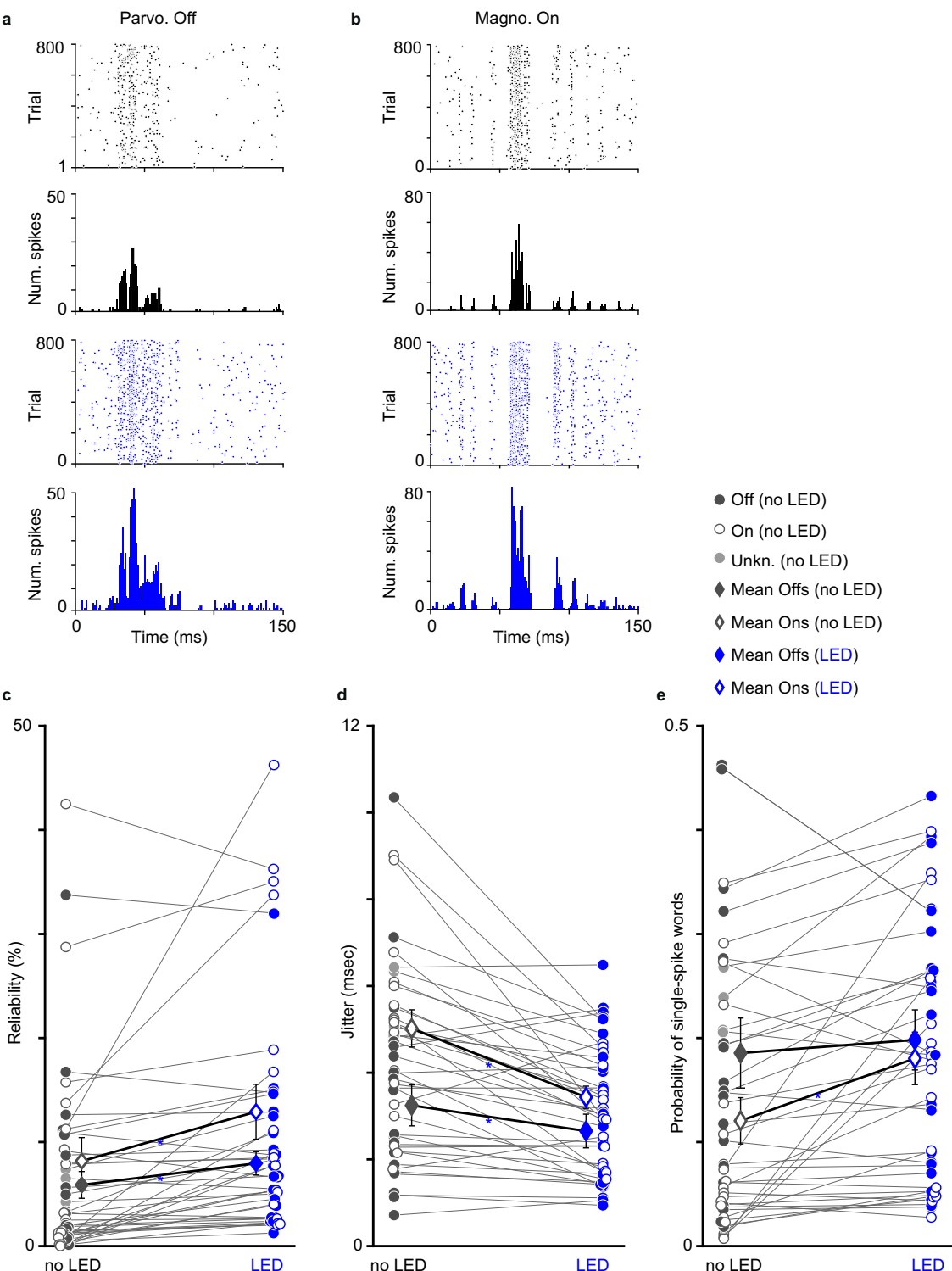

**Fig. 7 | Optogenetic modulation of LGN neuronal spike timing. a** Rasters and PSTHs for a representative parvocellular Off LGN neuron showing responses to the flashing black spot without (top, black) and with (bottom, blue) synchronized LED stimulation. Time = 0 is spot onset. **b** Rasters and PSTHs for a representative magnocellular On LGN neuron, conventions as in (**a**). **c** Spike timing reliability without (gray) and with (blue) LED stimulation for On (open) and Off (filled) LGN neurons. Neurons of unknown On/Off type indicated in light gray (see legend top right). Open and filled diamonds (interior) illustrate averages for On and Off LGN neurons per LED condition; error bars are SEMs, blue asterisks indicate significant differences (see Table 3 for statistics). **d** Spike timing jitter without (left) and with (right) LED stimulation; conventions as in (**c**). **e** The probability of single-spike words without (left) and with (right) LED stimulation; conventions as in (**c**, **d**). Source data are provided in Source Data file.

**Table 3 | Spike timing precision measures**

| | Reliability (%) No LED / with LED | | Jitter (msec) | | Single-spike word probability | | Latency (msec) | |
|---|---|---|---|---|---|---|---|---|
| All LGN neurons (n = 42) | 7.1±1.5 | 10.8±1.7 | 4.3±0.4 | 3.1±0.2 | 0.15±0.02 | 0.19±0.02 | 29.5±1.1 | 28.9±1.1 |
| LED effect: P values (paired t-test) | 0.0013 | | 1.8×10⁻⁵ p < 0.01 Wilcoxon | | 0.0038 | | 0.27 | |
| On neurons (n = 25) | 8.1±2.3 | 12.8±2.7 | 5±0.4 | 3.4±0.3 | 0.12±0.02 | 0.18±0.03 | 30.5±1.8 | 29.2±1.8 |
| LED effect: P values (paired t-test) | 0.01 | | 2.4×10⁻⁴ p < 0.01 Wilcoxon | | 0.0032 | | 0.16 | |
| Off neurons (n = 17) | 5.8±1.3 | 7.8±1.1 | 3.2±0.5 | 2.6±0.4 | 0.19±0.03 | 0.2±0.03 | 28±0.9 | 28.4±1 |
| LED effect: P values (paired t-test) | 0.0074 | | 0.0095 p < 0.01 Wilcoxon | | 0.47 | | 0.33 | |
| Parvo neurons (n = 14) | 4.5±1.3 | 7.4±1.4 | 3.4±0.6 | 2.3±0.3 | 0.18±0.04 | 0.21±0.04 | 28.1±1.9 | 27.9±1.9 |
| LED effect: P values (paired t-test) | 0.0005 | | 0.037 | | 0.09 | | 0.19 | |
| Magno neurons (n = 12) | 2.7±1.2 | 6.3±1.5 | 4.2±0.7 | 2.7±0.3 | 0.1±0.03 | 0.16±0.03 | 29.1±2.4 | 27.8±2.5 |
| LED effect: P values (paired t-test) | 0.046 | | 0.0073 p < 0.01 Wilcoxon | | 0.09 | | 0.28 | |
| Konio neurons (n = 5) | 7.1±3.4 | 8.5±1.9 | 4.5±1.1 | 3.9±0.9 | 0.2±0.09 | 0.3±0.05 | 33±3.7 | 29.8±3.6 |
| LED effect: P values (paired t-test) | 0.47 | | 0.32 | | 0.44 | | 0.37 | |

Average reliability, jitter, probability of single-spike words, and response latency for all LGN neurons, for On and Off neurons separately, and for identified parvocellular (parvo), magnocellular (magno), and koniocellular (Konio) neurons; for each metric average, values without LED at left, with LED at right (LED labels only shown for reliability columns). P-values indicate significant effects across LED conditions (paired t-tests, two-sided), using a Bonferroni corrected alpha = 0.0125.

viral genes[58]. Injection of SADΔG-ChR2-mCherry into the LGN caused expression of ChR2 and mCherry in retinogeniculate neurons, thalamic reticular nucleus (TRN) neurons[60], LGN interneurons Fig. 1bi, and CG neurons Fig. 1c since these populations have axon terminals in the LGN. Importantly, because CG neurons are the only visual cortical neurons with axon terminals in the LGN[36–39,61], they are the only neurons in the cortex expressing ChR2 and mCherry following injection of virus into the LGN[22,41,62,63]. All three monkeys used in this study underwent virus injections in sterile recovery surgery followed by a terminal neurophysiology recording experiment, the endpoint of which was euthanasia to enable histological processing of brain and eye tissue. In one monkey (Monkey L), a second version of the G-deleted rabies virus containing genes encoding the inhibitory opsin ArchT and GFP (SADΔG-ArchT-GFP, titer: 4.1 × 10⁹, provided by Callaway lab) was injected into the LGN in the opposite hemisphere from the ChR2/mCherry virus injection.

For a cross-species comparison of effects of CG feedback on the spatial resolution of LGN neurons, we also analyzed LGN neuronal responses from five female ferrets (*Mustela putorius furo*. Marshall Farms) aged 1-2 years. All procedures involving ferrets conformed to the guidelines set forth by the NIH and the USDA and were approved by the Institutional Animal Care and Use Committee at the University of Rochester. Surgical preparation and neurophysiological data collection involving ferrets has been described in detail previously[22,41,62–64]. All five ferrets had successful injections of SADΔG-ChR2-mCherry (same titer range and source as above) into the LGN as verified histologically post-mortem. Visual and optogenetic stimulation as well as neurophysiological data collection and analyses in ferrets were the same as described below.

**Surgical procedures for virus injection**

Recovery surgeries were conducted in a sterile surgical suite using aseptic techniques. Monkeys were initially anesthetized with ketamine (10 mg/kg, IM) sometimes with xylazine (0.2–2 mg/kg, IM), or midazolam (0.05–2 mg/kg, IM), then intubated and maintained under full surgical anesthesia with isoflurane (1–3% inhaled in oxygen). Animals were placed in a stereotaxic frame, wrapped in a thermostatically controlled heating blanket, and prepped for surgery. Throughout surgical procedures, animals were continuously monitored for heart rate, respiration rate, expired $CO_2$, SPO₂, and temperature. Monkeys received a continuous infusion of lactated ringers (5-10 ml/kg/hour, IV) to prevent dehydration and were also given atropine (0.05 mg/kg, IM) to reduce mucus production and dexamethasone (1–2 ml/kg, IM) to reduce inflammation. Cefazolin (22 mg/kg/hour, IV) was administered throughout the surgery as an antibiotic.

A midline scalp incision was made and the muscles retracted. A small (~8mm²) craniotomy was made in one hemisphere located dorsally and according to stereotaxic coordinates for parafoveal regions of the LGN (~AP = + 5.5, ML = 11). A sharp platinum/iridium recording electrode (FHC, Bowdoin, ME) was inserted through the dura and lowered into the brain and neuronal responses to light flashes in the eyes were used to determine the location, depth, and span through the LGN. The recording electrode was removed and an injection needle attached to a Hamilton syringe was placed in the same stereotaxic location identified from recordings and lowered to the appropriate depth (~27–30 mm). Virus injections were made at multiple depths, spanning all 6 layers of the LGN, by pressure applied to the Hamilton syringe. Between 3 and 10 microliters of virus was injected into the LGN in total over 1 or 2 separate injection needle penetrations. Injected virus did not fill the entire LGN, but covered similar proportions of caudal LGN in each animal, spanning all layers of the LGN, and corresponding to parafoveal eccentricities (Fig. 1b). The injection needle was removed and gel foam or bone wax was placed over the craniotomy and the muscles and skin were sutured together. Animals were given buprenorphine (0.01 mg/kg, IM) and meloxicam

(0.2 mg/kg, IM) for analgesia pre- and post-operatively, and recovered for 7 days to allow virus-mediated expression of opsins and fluorescent markers, after which the terminal neurophysiological recording session began.

## Surgical procedures for neurophysiological recordings

For anesthetized recording procedures, monkeys were initially anesthetized with ketamine (10 mg/kg, IM) and/or xylazine (0.2–2 mg/kg, IM), then intubated and maintained with sufentanil citrate (8–24micrograms/kg/hour, IV) or fentanyl (10-25micrograms/kg/hour, IV), isoflurane (0.25–1.5%), and nitrous oxide (1:2, in oxygen). Animals were placed in a stereotaxic frame and wrapped in a thermostatically controlled heating blanket. Throughout the procedures, animals were continuously monitored for heart rate, respiration rate, expired $CO_2$, $SPO_2$, temperature, and EEG. Blood pressure was also measured invasively or non-invasively (continuously or intermittently, respectively). Proper anesthetic plane was also assessed by monitoring the subcranial EEG for changes in slow-wave/spindle activity in addition to monitoring heart rate, blood pressure, and expired $CO_2$. If changes in any of these physiological measures indicated a decreased level of anesthesia, additional sufentanil/fentanyl was given and the rate of infusion increased. Animals received a continuous infusion of lactated Ringer's (5–20 ml/kg/hour, IV), often with dextrose, to prevent dehydration, and cefazolin (11 mg/kg/hour, IV) as an antibiotic. Atropine (0.05 mg/kg, IM) and dexamethasone (1–4 ml/kg, IM) were also administered at the start of the surgical procedures. A urinary catheter was placed and urine collected and measured daily. Blood gases and blood glucose were measured daily and Ringer's solutions adjusted in electrolyte and/or dextrose content accordingly.

The prior scalp incision was reopened and extended. A craniotomy was made over V1 and the dura was removed over electrode insertion sites in both LGN and V1 craniotomies. Craniotomies were filled with agar (1% in sterile saline). The eyes were fitted with contact lenses and focused on a tangent screen located ~60 cm in front of the animal. In some cases, prior to placing contact lenses, a retinoscope was used to estimate refractive errors. Corrective external lenses were then also placed ~3 cm in front of each eye to focus stimuli on the monitor. Once all surgical procedures were complete there was a 30-minute physiological monitoring phase, after which animals were paralyzed with vecuronium bromide or rocuronium bromide (0.1–0.6 mg/kg/hour, IV) to eliminate eye movements.

After neurophysiological recordings were complete (100 hours after initial anesthesia induction), animals were euthanized with an overdose of sodium pentobarbital (80 mg/kg, IV) and perfused transcardially with cold phosphate buffered saline (PBS) followed by 4% paraformaldehyde in 0.1 M phosphate buffer (PB). Brains were removed and placed in 20% sucrose (sometimes with 4% paraformaldehyde) in PB. Sectioning and histology were performed ~5 days after the perfusion (after brains sunk in sucrose solution). Eyes were removed and retinas dissected and mounted on glass slides to confirm the presence and location of mCherry- or GFP-labeled retinal ganglion cells.

## Neurophysiological recordings

Extracellular spikes and local field potentials (LFPs) were recorded using two multi-electrode arrays: a 7-channel Eckhorn Matrix of independently movable platinum-tungsten microelectrodes (Thomas Recording GMBH, Giessen, Germany) inserted into the LGN; and a 24-contact U/V/S-probe (Plexon Inc., Dallas, TX) placed perpendicular to the opercular surface in V1 such that electrodes spanned all 6 cortical layers. The V1 and LGN arrays were placed in retinotopically aligned regions of V1 and LGN (Supplementary Fig. 1), assessed using standard hand- and software-assisted receptive field mapping procedures. Receptive field centers of simultaneously recorded LGN and V1 neurons were 0.47±0.11 degrees apart on average. Continuous voltage

signals from both arrays were amplified and digitized at 10,000 Hz using an Omniplex data acquisition system (Plexon Inc., Dallas, TX). Continuous voltage recordings from each electrode/contact were low-pass filtered (at 200 Hz) and down-sampled (to 1000 Hz) to extract LFPs; the same continuous voltage signals were also thresholded and high-pass filtered (at 300 Hz) to extract spiking activity. Timestamps corresponding to the onset of visual stimuli and triggers of LED stimulation were also recorded by the Omniplex system. Recordings of LGN and V1 neurons were made in response to presentations of LED stimulation alone, visual stimuli alone, and visual stimuli synchronized with LED stimulation.

## Optogenetic stimulation

A blue LED emitting 465 nm light (Doric Lenses Inc., Quebec, CAN or PlexBright by Plexon Inc., Dallas, TX) coupled to a fiber optic cable (200 μm, NA:0.53 or 200 μm, NA: 0.66) was embedded in the agar touching the surface of V1 immediately adjacent to the electrode array in V1. Light intensity at the tip of the fiber measured between 20 and 90 mW/mm²; light intensity at layer 6 was estimated to be 0.2 to 0.9 mW/mm[265]. To activate ArchT, a green LED emitting 525 nm light was used with light intensity at the tip around 20 mW/mm². The spatial spread of the LED was estimated to correspond to 1–2 functional hyper-columns in parafoveal, opercular V1[22,66]. Neuronal recordings were made during LED stimulation alone using three different LED pulse protocols: 1) the LED was on for 20 msec, off for 30 msec, alternating 5 times then pausing for the remainder of 1 second, and this pattern was repeated at least 10 times; 2) the LED was on for 200 msec, off for 800 msec, and this pattern was repeated at least 10 times; and 3) the LED was on for 5 msec and pulsed over 2 seconds at a frequency varying between 0.5 Hz and 16 Hz in steps of 10, and this pattern was repeated at least 2 times. The third LED stimulation protocol was used in all three monkeys and the first and second protocols were also used for Monkeys C and K to increase the likelihood of detecting LED-only responsive neurons.

Neuronal recordings were also made while LED stimulation was synchronized to displayed visual stimuli, described in the next section (synchronization achieved by a TTL pulse transmitted from the visual stimulus generation system to the LED driver). Synchronization of LED pulses per visual stimulus were as follows: 1) for drifting grating stimuli, the LED was triggered once (5 msec duration per triggered pulse) at the start of each grating cycle, i.e. the LED flashed at the temporal frequency of the drifting grating and was precisely aligned to the grating phase; 2) for the flashing black spot stimulus, the LED was triggered once (5 msec duration per triggered pulse) at the onset of each spot presentation, i.e. the LED flashed at 4 Hz precisely aligned with spot onset; 3) for the m-sequence stimulus, the LED was triggered at 4 Hz (5 msec duration per triggered pulse) for the duration of the stimulus display, starting at the presentation of the first frame of the stimulus. Neuronal responses measured during LED stimulation alone were quantified by computing peri-stimulus time histograms (PSTHs) and rasters in which the "stimulus" was the LED onset and bin widths were 5 msec.

## Visual stimulation

Gray-scale drifting sinusoidal gratings, flashing black spots, black/white m-sequence white noise stimuli, and cone-modulating m-sequence stimuli[67] were generated using a ViSaGe system (Cambridge Research Systems, Rochester, UK) and presented on a gamma-calibrated CRT monitor (ViewSonic, Brea, CA) with a refresh rate of 100 Hz, resolution of 800×600, and a mean luminance of 38 candelas/m² that was positioned ~60 cm in front of animals' eyes. The monitor was the only source of illumination near the animal. All stimuli were generated with custom-written Matlab (Mathworks, Natick, MA) command scripts. All stimuli were centered at the receptive field locations of recorded neurons, as measured using hand- and software-

assisted mapping techniques. Gratings and spots were between 0.25 and 5 degrees in diameter (except when varying in size). In many instances, a single stimulus (grating or spot) overlapped multiple proximal/overlapping receptive fields of simultaneously recorded neurons. In some instances when recorded neuronal receptive fields were more spatially displayed, multiple gratings or spots (identical in size and other parameters) were displayed simultaneously, each centered on the corresponding receptive fields of recorded neurons. Gratings and flashing spots were presented for two seconds followed by two seconds of mean gray. Gratings varied in contrast (1–100%), temporal frequency (1–32 Hz), spatial frequency (0.2–1.5 cycles/degree), orientation (0–324 degrees), or size (0.3–12 degrees in diameter) in 10 steps. When not varying, grating parameters were fixed at the preferred spatial and temporal frequencies of the majority of recorded LGN neurons, preferred orientation of recorded V1 neurons, a contrast of 70%, and the preferred size and/or size just large enough to cover all nearby receptive fields. All grating stimuli were each displayed a minimum of 4 times, at least twice synchronized with LED stimulation of CG feedback and at least twice without LED stimulation. The flashing black spot was displayed for 30 msec at 4 Hz (over 2 seconds), and this pattern was repeated a minimum of 40 times (minimum 320 flashes), half of the trials with LED stimulation of CG feedback and half without LED stimulation. M-sequence stimuli were 10-14 degrees on each side of a square 16 × 16 pixel grid and were displayed for ~20 minutes on each of two repeats, once with and once without LED stimulation of CG feedback. The luminance/color of each pixel (black/white or L/M/S-cone On/Off RGB values) modulated according to the m-sequence every two frames (see Fig. 6a, b insets).

## Neuronal identification and tuning measurements

Single unit spikes recorded on all channels were sorted offline using commercial spike sorting software (Offline Sorter by Plexon Inc., Dallas, TX) employing principal components analysis (PCA). From 32 total LGN recording penetrations (15 from Monkey L, 12 from Monkey K, 5 from Monkey C, with between 4 and 7 operating electrodes per penetration), a total of 141 well isolated single units were identified from 185 total possible recording sites (for hit rates of 73, 75, and 77% per monkey). These 141 single units were defined by meeting the following criteria: 1) spike waveforms generated well-isolated clusters in the PCA space; 2) short inter-spike interval (ISI) violations (ISIs<1msec) were present for <0.1% of sorted spikes; 3) signal-to-noise ratios (SNRs) for sorted spike waveform shapes were >2.75[68]; and 4) the same spike waveform could be tracked on the same electrode across all recordings made during that penetration. A further visual responsiveness criterion was imposed for inclusion in the analyses for this study: single units needed to demonstrate significant tuning for drifting gratings varying in contrast, spatial frequency, and temporal frequency (determined by ANOVA) and/or show defined spatial receptive fields based on m-sequence responses and/or show significant flashing spot responses (determined by reliability calculation, described below). Application of this visual responsiveness criterion reduced the total sample to 98 visually responsive/tuned LGN neurons across the three monkeys. From Monkey L, 21 LGN neurons from the hemisphere with ChR2 expression (3 were defined as parvocellular, 9 as magnocellular, and 3 as koniocellular neurons, as described below, and 6 had responses only to flashed spots, so were of unknown type). Also from Monkey L, 10 LGN neurons were from the hemisphere with ArchT expression (3 parvocellular and 7 magnocellular neurons). From Monkey C, 23 LGN neurons were from one hemisphere with ChR2 expression (8 parvocellular, 9 magnocellular, and 1 koniocellular neuron and 5 were unknown due to flash-only responses). From Monkey K, 44 LGN neurons were from one hemisphere with ChR2 expression (19 parvocellular, 13 magnocellular, and 12 koniocellular neurons). LGN neurons of the same type recorded across different monkeys demonstrated similar tuning across multiple tuning

dimensions (Supplementary Fig. 2), so neurons recorded across all monkeys were pooled together.

For analyses of ferret LGN neurons during optogenetic manipulation of CG feedback, 33 LGN neurons were included from 5 ferrets, in one hemisphere of LGN each in which there was ChR2 expression (12 Y and 21 X neurons in total). No differences were observed across ferrets in tuning properties among neurons of the same type, so data were pooled across ferrets.

For this study, V1 neurons in monkeys were only analyzed for responses to LED stimulation alone to identify putative CG neurons. The approximate laminar positions of neurons recorded in V1 were estimated from current-source density (CSD) profiles generated from visually evoked LFPs recorded across the cortical layers in response to flashed black spots. The boundary between layer 4C and 5, corresponding to the first polarity reversal in visually evoked LFPs and CSDs[22,69,70], was used toward identifying putative CG neurons. Specifically, putative CG neurons in V1 were defined based on three criteria: 1) single units had to be well-isolated (as described above); 2) single units had to be located at least 3 contacts (>200 microns) below the layer 4C/5 border defined by the LFPs and CSD spectra; and 3) single units had to respond to LED-only stimulation within 5 msec of LED onset with high reliability, low latency jitter, and spike counts within the 20 msec window following LED onset that were at least 2 standard deviations above spontaneous spike counts measured 50 msec before LED stimulation[22]. We then computed hit rates for identifying putative CG neurons from the number of well-isolated single units we could record on deep layer contacts per penetration. In 3 V1 penetrations per monkey, well-isolated deep-layer V1 neurons were identified. About 5 contacts were estimated to be in layer 6 per penetration and the proportion of CG neurons in layer 6 is about 15%[36], yielding a "best-case" likelihood of encountering ~2 putative CG neurons per monkey. In Monkey L, we identified 2 putative CG neurons (equal to best-case hit rate), in Monkey C we identified 4 putative CG neurons (double best-case hit rate), and in Monkey K we identified 7 putative CG neurons (3.5 times best-case hit rate; see Fig. 1d for examples per monkey). Interestingly, we also identified 8 LGN neurons (1 in Monkey L, 3 in Monkey C, and 4 in Monkey K) that were also responsive to LED stimulation alone, using the same criteria applied to V1 neurons except the response onset could be within 20 msec of LED onset (see Fig. 1e for example). Accordingly, the proportions of recorded LGN neurons responsive to LED stimulation alone were 5% (Monkey L), 13% (Monkey C) and 16% (Monkey K).

All subsequent analyses were performed on LGN neurons. Quantifications of neuronal responses to visual stimuli and LED stimulation were performed using custom-written Matlab scripts. All tuning responses, curve fits, tuning metrics, and other physiological measures were calculated separately for the visual stimulus alone condition and the visual stimulus synchronized with LED stimulation condition. Neuronal tuning was assessed by calculating the average firing rate in response to repeated presentations of drifting sinusoidal gratings varying in a given parameter (contrast, spatial frequency, temporal frequency, and size). Average tuning responses were then fit with functions to derive tuning curves, from which tuning metrics were computed. For contrast tuning data, neuronal responses were fit with a Naka-Rushton function[71], from which we computed the c50, or the contrast to evoke a half-maximum response. For spatial frequency (SF) and temporal frequency (TF) tuning data, neuronal responses were fit with a difference of Gaussians function and the preferred SF or TF, i.e., the frequency to evoke the maximum response, was determined. From TF curves, a second metric, the TF high50, or the TF to evoke a half-maximal response on the falling (high-TF) side of the tuning curve, was also computed. Neuronal responses to gratings increasing in size (grating diameter) were also fit with a difference of Gaussian function and the preferred size determined from the maximum response. In addition, a surround suppression index (SSI) was computed from each

curve as the difference divided by the sum of the response to the preferred size and the response to the largest size gratings, both baseline-subtracted.

Additional neurophysiological measures assessed per LGN neuron included average spontaneous firing rate, measured in between stimulus displays while the CRT monitor displayed mean gray, and average visually evoked firing rate calculated as the average response to high-contrast gratings of preferred spatial and temporal frequency and preferred size. To quantify gain effects of LED stimulation of CG feedback on neuronal tuning responses, we computed area under the curve (AUC) values (baseline-subtracted) for spatial and temporal frequency tuning curves, again computed separately for with and without synchronized LED stimulation conditions.

LGN neuronal responses to m-sequence stimuli were analyzed using standard reverse correlation and spike-triggered averaging (STA) methods, as in ref. 22. Spatiotemporal receptive fields generated from luminance-modulating and cone-modulating stimuli were compared to assist in LGN neuron type classification. Receptive field area was computed from the peak response frame of each well-defined STA that was not near the stimulus edge. Each receptive field was fit with a 2D Gaussian and the number of pixels with luminance values 6-8 standard deviations above (or below for Off STAs; criterion always matched across conditions) the mean luminance value was calculated and then converted into degrees based on the view distance to the monitor and the stimulus size. Additionally, the temporal modulation in the luminance of the brightest pixel in the receptive field was measured to facilitate latency calculations from white noise responses (described below).

LGN neurons were defined as parvocellular, magnocellular, or koniocellular neurons based on multiple observations, including estimates of the recording layer/depth, ipsilateral or contralateral eye input, combined tuning metrics, and luminance and cone-modulating STAs. Subjective classifications were straightforward, given these observations. For example, LGN neurons with clear opponent S- and L/M-cone STAs from cone-modulating m-sequence stimuli were defined as koniocellular neurons, while LGN neurons with clear opponent L- and M-cone STAs were defined as parvocellular neurons. To independently verify LGN neuron type classification using an unbiased method, we performed a cluster analysis on 48 LGN neurons with significant tuning for all grating stimuli using the following tuning metrics (all from the visual stimulus alone, i.e. no LED, condition): c50, preferred TF or TF high50, and SSI (Supplementary Fig. 2). In the resulting clusters, 8 neurons were clustered differently relative to our subjective classification. We performed similar subjective classification and independent unbiased clustering to ferret LGN neurons using the same tuning metrics and only 3 of 22 ferret LGN neurons were clustered differently relative to our subjective classification.

For LGN neurons with responses to flashed spots and/or m-sequence stimuli, we defined neurons as On or Off based on the latency of the maximum response to the flashed black spot (latencies <50 msec after black spot onset were Off, latencies >50 msec after black spot onset were On) and/or the luminance polarity of the STA. In no cases were flashing spot and m-sequence responses in conflict in terms of defining On or Off type.

### Latency and reliability measurements
Effects of optogenetic stimulation of CG feedback on response latencies and spike timing precision were examined from the responses of 42 LGN neurons (5 from Monkey L, 9 from Monkey C, 28 from Monkey K) to flashing black spots with and without synchronized LED stimulation. Response latencies were calculated as the time from spot onset (for Off neurons) or spot offset (for On neurons) to the half-maximum response bin in the PSTH (1msec bins). Response latencies were also computed from brightest pixel temporal luminance modulation

curves from white noise STAs as the time between the peak/trough in the curve and time = 0, or the time of the spike.

Initially, we computed spike timing precision as the standard deviation of the first spike time following flashed spot onset/offset[72] as we had done previously for ferret LGN neurons[22]. Although LED stimulation reduced the standard deviation of the first spike time across monkey LGN neurons ($p = 5.05 \times 10^{-5}$, paired t-test), we wanted to further assess spike timing precision using measurements of reliability and jitter[73]. Reliability was computed as the probability of a spike occurring in the 1-msec bin corresponding to the maximum response bin in the PSTH. Probabilities were then converted into percentages. Jitter was computed as the sigma of a Gaussian fit to bins in the PSTH ±2-5 msec around the maximum response bin.

Because many LGN neuronal responses to flashing black spots included more than one clear peak in the PSTH and response patterns appeared somewhat stereotyped across trials, we performed an analysis of spiking patterns (or "words") in a 10-msec window starting 1 msec prior to the peak bin[74]. Within this 10-msec window, we computed the distribution of words (all words were 10 msec in length with 1-msec bins) across trials per condition. From each distribution, we extracted the probability of single-spike words and the probability of words with two or more spikes. For each LGN neuron, overall word distributions and probabilities of single-spike words and of >2-spike words were then compared across conditions with or without synchronized LED stimulation.

### Statistical analyses
To compare tuning metrics, spontaneous and stimulus-evoked firing rates, and AUC measures across different types of LGN neurons (parvocellular, magnocellular, or koniocellular), we utilized non-parametric ANOVAs (Kruskal-Wallis tests) using alpha = 0.05. All comparisons across LED conditions (i.e., visual stimuli alone versus visual stimuli synchronized with LED stimulation) were made using paired t-tests for normally distributed measures and supplemented with Wilcoxon signed-rank tests for non-normally distributed measures. Comparisons across LED conditions were made for each tuning metric, receptive field size, spontaneous and stimulus-evoked firing rates, and both AUC measures. All of these comparisons were made separately for LGN neurons of each type. We applied a Bonferroni correction of 4 to account for 4 possible comparisons per neuron for contrast, SF, and TF tuning metrics, for a corrected alpha = 0.0125. Because we did not test neuronal responses at spatial frequencies higher than 1.5 cycles/degree, we excluded from the statistical comparison of preferred spatial frequency across LED conditions all LGN neurons that did not reach peak firing rate for a spatial frequency below 1.5 cycles/degree. This excluded 7 parvocellular and 1 magnocellular neuron from these t-tests. A slightly different sample was used for comparisons of size tuning metrics across LED conditions, so for comparisons of preferred grating size and SSI, we applied a Bonferroni correction of 2 with a corrected alpha = 0.025. Importantly, size tuning metrics (SSI and preferred grating size) were significantly different across LED conditions for magnocellular neurons recorded within each monkey individually, in addition to being significantly different for the pooled sample.

For statistical comparisons of response latency, reliability, jitter, and word probabilities, different groupings of LGN neurons were utilized. First, because prior results for ferret LGN neurons suggested that all LGN neurons may show similar effects of LED stimulation on these measures[22], we pooled all LGN neurons together and compared latency, reliability, jitter, and word probabilities across LED conditions using a paired t-test for normally distributed measures and supplemented with Wilcoxon signed-rank tests for non-normally distributed measures. Next, given the fact that the flashing black spot could differentially impact On and Off LGN neurons, we separated On and Off LGN neurons and again computed the same measures across LED

stimulation conditions. Finally, we also separated LGN neurons that we could define as parvocellular, magnocellular, or koniocellular and again computed the same measures across LED conditions. For each of these statistical comparisons, we applied a Bonferroni correction of 4 to account for 4 possible comparisons per neuron (tests of: all, On, Off, neuronal type) for a corrected alpha=0.0125.

## Staining

Brain tissue from all monkeys was sectioned coronally at a thickness of 70 microns per section using a freezing microtome (Thermo Scientific, Waltham, MA). All sections were first stained for cytochrome oxidase activity in order to visualize cortical layers and subcortical nuclei. All sections were then labeled with a primary antibody against mCherry or GFP (rabbit anti-DS red [1:1000 dilution], Clontech Laboratories Inc., Mountain View, CA [catalog #632496] or rabbit anti-GFP [1:1000 dilution], Molecular Probes/Life Technologies, Grand Island, NY [catalog #6A-11122]), followed by a biotinylated secondary antibody (biotinylated goat anti-rabbit [1:500 dilution], Molecular Probes/Life Technologies, Grand Island, NY [catalog #31820]) before reacting with DAB/peroxidase such that all labeled neurons were permanently stained. Sections were then mounted on glass slides, defatted, and cover slipped. Sections were later viewed and photographed using a microscope (Nikon Instruments Inc., Melville, NY) with an attached Optronics camera to verify the expression of fluorescent markers in injected hemispheres. The relative proportions of virus-labeled CG neurons in lower versus upper layer 6 were computed per monkey by counting labeled CG neurons in 10 unique, randomly selected sections in which a minimum of 5 labeled CG neurons were visible within a 5X objective view. Retinas were dissected, flat-mounted onto glass slides, and cover slipped to enable direct fluorescence detection using the microscope.

## Reporting summary

Further information on research design is available in the Nature Portfolio Reporting Summary linked to this article.

## Data availability

Data generated for this study are currently being formatted for deposit into the NDI repository. In the meantime, data may be made available upon request to the corresponding author. Source data for figures and tables are provided with this paper. Source data are provided with this paper.

## Code availability

The code is available at https://github.com/BriggsNeuro/Tuning-and-opto.

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

## Acknowledgements

We thank Marc Mancarella, Brianna Carr, and Elise Bragg for their expert technical assistance and Drs. Dana LeMoine, Katherine Nolan, Diane Moorman-White, Jeff Wyatt, Karen Moodie, and Kirk Maurer for veterinary assistance. We thank Dr. Jacqueline Hembrook-Short, Dr. Vanessa Mock, Silei Zhu, and Jingyi Yang for assisting with data collection. We thank Dr. Peter Lennie for helpful discussions of data and Drs. Laura Busse and Marty Usrey for helpful discussions of the manuscript. This work was funded by the National Institutes of Health (NEI: R00 EY018683 and R01 EY025219 to F.B., T32EY007125 and F31EY032332 to A.J.M., and a Center for Visual Science Instrumentation Core grant: P30EY001319; and NINDS: U01 NS131810) and the Whitehall Foundation (2013-05-06). S.M. was supported by a Center for Visual Science Summer Fellowship (through NIH T32EY007125), a Meliora Fellowships from the Brain and Cognitive Sciences Department, and a Discover Grant from the University of Rochester Office of Undergraduate Research. J.M.H. was supported by a Graduate Fellowship from the Albert J. Ryan Foundation.

## Author contributions

A.J.M, J.M.H., and F.B. designed the experiments. A.J.M., J.M.H., and F.B. collected the data. S.M., A.J.M., and F.B. analyzed the data. S.M., A.J.M., and F.B. wrote the manuscript.

## Funding

## Competing interests

The authors declare no competing interests.
