## [Transparent Peer Review file · Nature Communications]

Dual parallel stream-specific and generalized effects of corticogeniculate feedback on LGN neurons in primate and carnivore

Corresponding Author: Dr Farran Briggs

Version 0:

Reviewer comments:

Reviewer #1

(Remarks to the Author)

Mai et al very elegantly demonstrate in monkeys and ferrets the effect of V1 cortical feedback on LGN neuron receptive field properties. Previous research has relied on classical non-selective approaches, such as pharmacological inactivation or cooling of V1. Here, the application of cutting edge retrograde optogenetic techniques allows for a specific targeting of V1 cortico-geniculate (CG) neurons projecting as feedback to LGN. Applying optogenetic methods in primates has proven to be a major challenge and therefore the new findings provide important conceptual and methodological advancement. The authors findings move the field ahead towards much needed cell specific pathway mapping in primate circuits.

My only major concern relates to the specificity of the optogenetic technique and how it might have contributed to the specific magnocellular effects on LGN receptive field (RF) size and surround suppression. Previous research has demonstrated such effect also for the parvocellular system (Jones et al, J Neurosci, 2012). However, such effects might be smaller for parvo in comparison to magno (Alitto & Usrey, Neuron, 2008) and based on their timing inconsistent with CG feedback effects. I believe a more extensive demonstration and if possible, quantification of the opto construct expression would be desirable. I will try to make a few specific suggestions for the authors' consideration.

1. Figure 1Bi and Bii nicely show LGN expression in monkeys L and C. In comparison the expression appears much weaker in monkey K (Biii). Moreover the expression at least in the displayed sections seems primarily focused on parvocellular, and not so much magnocellular layers. With this expression pattern the absence of parvocellular RF size effects seem surprising. Could you please comment on the strength of expression magno vs parvocellular, and how it might affect the electrophysiological results?

2. Briggs (Neuron, 2026) and Fitzpatrick (Vis Neurosci, 1994) report a bimodal expression of parvocellular vs magnocellular projecting CG neurons in layer 6 the upper and lower third of layer 6, respectively. From Figure 1C it seems that this is also the case here. Again, with this expression pattern the absence of parvocellular RF size effects seem surprising, and the authors may want to comment further.

3. From Figure 1Bi expression in magnocellular neuron cell bodies seems a possibility. If this is the case this might open up the possibility of antidromic activation of LGN axons to V1, at least in theory. In my experience, such imperfections of virus expression are usually present, though rather minor, in comparison to the principal mechanism. Nevertheless, if antidromic activation remains a minor possibility, the authors may want to opt for more cautionary wording.

4. It would be good get more detail about the optogenetically identified CG neurons, including whether they are visually driven and to what extent their receptive field (RF) overlaps with the sampled LGN neurons. The degree of RF overlap is known to be a critical variable and excitatory vs inhibitory effects have been reported, at least for parvocellular neurons (Jones et al, J Neurosci, 2012).

Minor:

5. I appreciate different reporting styles and do not want to criticize the authors. From my perspective, I appreciate explicit reports on negative effects in the abstract, such as the suspected lack of surround suppression in parvocellular neurons.

6. For experiments that apply both visual as well as optogenetic stimulation, the authors refer to synchronization between

LED and visual stimulation. Could you please provide additional details how this synchronization was achieved and maintained throughout varying stimulus conditions and cell classes?

7. I would appreciate defining the m-sequence stimulus more clearly. Including a schematic graphic of the stimulus in different conditions (achromatic, cone-modulating separately) could be very helpful in comprehensive understanding of the experimental design, allowing for better visualization and interpretation of the stimulus effects.

8. Consider adding an indication of significance to Figure 3. The figure appears complex, with abundant information. The arrows on the plots might be misleading, as they point at both trends and significant differences. To enhance clarity, bar plots showing average results for each cell type could be included.

9. In Figures 4, 5D, and 6, there is also no clear indication of significance. Arrows are also present in plots where the difference is not significant or nonexistent (4C, 5D). Consider either:

- o Removing arrows in all plots where the difference is not significant.
- o Adding a star or other indicative symbol to highlight significant findings.

Reviewer #2

(Remarks to the Author)

This is the first study, to my knowledge, using optogenetic activation and inactivation of corticogeniculate feedback (CG, V1 to LGN) to examine its role in LGN information processing and coding. They discovered stream-specific effects of CG feedback in improving the spatial resolution of magnocellular LGN neurons, and found that enhancing CG feedback increased extraclassical surround suppression, shrunked the size of LGN receptive fields, and increased preferred spatial frequencies of magnocellular LGN neurons. In turn, optogenetically suppressing CG feedback reduced surround suppression and increased preferred stimulus size. They further conducted cross-species comparisons (anesthetized monkeys and ferrets) to find that enhancing CG feedback reveals similar stream-specific effects in LGN Y neurons, and that optogenetically enhancing CG feedback improved LGN response reliability and reduced response jitter across neuronal types. Overall, this study is timely and reports some interesting findings although I'm unsure about the general relevance for the broad readership of Nature Communications. In particular, they seem to report results that are not necessarily connected. In addition, the cell count is very small, and they lack multiple comparison statistics. So I'm not fully convinced of the validity and robustness of the results, as presented here, and how exactly they will advance the field.

Specific comments:

1) The paper is not well written, especially considering the broad readership they target. As presented, the results appear as a bunch of findings that are not coherently presented into a story, many of the analyses appear ad-hoc, and the figures are not effective or self explanatory. Also, the paper is not written in the NPG format, for instance Materials and Methods appear immediately after Introduction. The statistics are presented in a table, but I feel they should have been part of the main text or the figure captions. I found it puzzling when they present the scientific premise that they don't cite the relevant papers in the very field they examine. There have been several papers already manipulating cortical feedback optogenetically (although not CG feedback), but these studies are not mentioned. In particular, a relevant study from Angelucci's lab (Nurminen et al., Nat Commun, 2018), in which the authors manipulated cortical feedback from V2 to V1 to alter responses and surround suppression is not even mentioned. I think that study is very relevant since it is possible that some of the surround suppression results the authors report here could originate from V2.

2) I am somewhat puzzled by the low number of cells across the studies, in the 20-30 range across 3 animals. True, they had many conditions they tested and controls, but they nonetheless used anesthetized animals and could have easily doubled the cell count. The main issue is the robustness of their results and more cells would have definitely increased confidence. In some of the analyses they used controls in which $n=4$, which is inappropriate. I would also like to see multiple comparison statistics, such as Bonferroni correction, applied to this data. They performed multiple tests on the same population of neurons, so this analysis is justified. Also, how many of their cells exhibit statistically significant effects? I couldn't find this information anywhere in the manuscript, but this is important knowledge that is currently missing.

3) They performed many analyses on the data, and even used a battery of stimuli to examine the role of CG feedback, however no justification is provided for that. Why spatial and temporal frequency, and why is temporal precision important in this context, and how is it related to the effects of surround suppression, and why is it important to study these effects across species? Currently, these manipulations and analyses appear simply thrown out to see what happens when probing the impact of CG feedback, but the coherent story is missing.

4) They conclude that the effects of CG feedback on the different metrics tested in LGN neurons are uncorrelated, but this is a bit surprising and the authors don't discuss the implications. Also, they present the magnocellular stream-specific effects of CG feedback on spatial receptive field properties as surprising, but this is actually expected. Overall, their results are not that surprising.

5) The figures are difficult to follow, and it is unclear whether they support the claims. They should clearly label individual examples and population effects, and indicate the statistical significance of the results, otherwise it is difficult to go back and forth to the text and the table to understand what's going on. They should also clearly label optogenetic activation and inactivation on each figure, not only in the text accompanying the figures.

Reviewer #3

(Remarks to the Author)

The observations here provide novel empirical data on a matter of strong interest – the impact of V1 projections on the activity of neurons in monkey LGN, the main visual input to V1. While this first feedback pathway in the visual system has been the subject of much previous work, in primates that work has had to rely on bulk inactivation of cortex by physical or chemical means. The current experiments use viral mediated optogenetic stimulation of cortical layer 6 neurons to provide much greater specificity. The measurements are technically challenging, requiring viral injection and precise electrophysiological measurements from a small and very deep thalamic brain structure, and successful optogenetic activation or suppression of neurons in cortex. An additional strength is that complementary measurements in ferret provide comparative data.

The main finding is that increasing cortical feedback changes spatial summation in Magno LGN pathway, alongside more general modulation of spike timing precision. The data is generally well described (though I think some of the aesthetics of the Figures could be improved), and the inferences are generally conservative. I therefore believe the experiments reported here are important addition to the literature.

I have some queries about particular aspects of the experiments and conclusions, below, but the main thought I had after reading was need for a clearer explanation of optogenetic effects on response amplitude. There are some rather large differences in the amplitudes of mean activity in some of the panels in Figure 2 – which I think are by and large not commented on, with focus on tuning. Better explanation of the impact of CG stimulation, for example, scatter plots of response amplitude in the LED / control conditions, might help the reader understand what is going on, but regardless the reader needs help here. On a related note, while there appear to be limited effects of optogenetic stimulation on average 'spontaneous' activity (Table 2), it is possible that optogenetic stimulation entrained activity without elevating or reducing it – it would be useful to see whether there was temporal modulation of firing rate at the rates of LED stimulation, in spontaneous or visually driven cases, where that is possible.

Inclusion of a descriptive but quantitative model (not one aimed at explaining the specific observations) may also help make the conclusions more coherent. In early parts of the Results you state that "Moreover, these effects are probably linked: by increasing extraclassical surround suppression, the classical receptive field area, and thus preferred stimulus size, shrink leading to an increase in the preferred grating spatial frequency. Thus, CG feedback is functionally stream-specific in the spatial domain, regulating receptive field size through extraclassical suppression for the magnocellular stream selectively". This mechanism would also explain the later results where there is increased gain but no change in receptive field size in the M-sequence experiments (the classical receptive field is unaffected, and a change in gain in the M-sequence experiments could be explained by reduced extraclassical suppression driven by the spatially extended flickering stimulus). If the extraclassical receptive field, including the cortical-driven component, is larger and more sensitive to lower spatial frequencies (which is likely) than is the classical receptive field, this would likely provide a sufficient explanation of the variety of spatial effects of stimulation.

1. Introduction: Paragraph 2 spends a good deal of time reiterating fairly standard description of roles of different thalamus groups (Parvo, Magno, Konio) however it feels unneeded here; would be good to focus this paragraph onto what the reader needs to know about CG feedback in particular.

2. Introduction: Paragraph 3 notes several different studies in various species on impact of causal manipulations of CG feedback. However the techniques used vary immensely and are many were fairly crude (no disrespect) disruptions, as is detailed in Discussion - it would be useful the reader is aware of that here (and thus contrasting the utility of the optogenetic approach).

3. Introduction: Paragraph 3: there is enough debate about the link between X/P and Y/M (as you make clearer in Discussion) that I am not sure it is necessary to be strong on the link here - just need to note that there are different pathways and different nomenclatures for different species.

4. Introduction: Paragraph 4: its not clear why visual stimuli that evoke unique responses across the parallel streams are important for understanding CG feedback. I may be missing something here, but its not spelled out.

5. Methods: 'Surgical procedures...' I think that both sufentanil and isoflurane were provided throughout the recordings, at level depending on physiological indicators, but would be good to make that explicit.

6. Methods: 'Optogenetic stimulation'. What was the rationale for adopting the three methods of stimulation? I think I can discern, but would be good to spell out. I am also remain somewhat unsure when each protocol was used, and would like to see this made clearer.

7. Methods: 'Visual stimulation'. "in steps of 10" -> "in 10 steps"?

8. Methods: 'Visual stimulation': How were the flashing spots positioned relative to receptive fields?

9. Methods: 'Neural identification and tuning measurements': Power law for contrast analyses: As I think a power law with exponent 2 only allows expansive relationships there is no saturation in contrast-response curves and c50 is largely undefined, so I am unsure how it is obtained. Also some of the fits in Figure 2, left column, look compressive not expansive. Am I missing something here?

10. Methods: 'Statistical analyses' Optogenetic activation is used in 3 hemispheres of 3 animals, and suppression used in 1 hemisphere of one animal. The low number of animals is to be expected in monkey experiments, and the authors have been able to obtain good numbers of LGN neurons in each animal – as in usual practice, most of the statistics pool observations across animals to investigate impact on individual neurons. However, a likely source of variance across neurons is the extent / expression of viral product in each animal, so it would be good to provide additional analyses which take animal into account. This may be particularly important in analyses where individual animals provide substantial majority of data (e.g. latency to flashing spots; Figure 6).

11. Results: it would be useful if Figure 1 or similar could show the overlap in receptive field locations of LGN and cortical neurons. As it stands, the only information I can find on the alignment of cortical stimulation/recording location, and LGN recording location, is the first sentence of paragraph 2 in results.
12. Results: there seems to be a missed chance to provide a more detailed description of the 8 LGN neurons directly responsive to cortical optogenetic activation. For example, were there any functional properties that might suggest particular class (e.g. interneurons, or Parvo/Magno/Konio).
13. Results: Figure 5 – a fairly strong text statement about ‘many LGN neurons had shorter spiking response latencies’ and ‘note blue curves are shifted towards 0’ in Results is at odds with the modest and even reversed timings in the Figure. Some clarification would be useful.
14. Results: it may be unlikely, but did optogenetic stimulation have an impact on cortical or LGN LFP power spectrum? That is, could reduced spike timing variability in LGN be related to disruption of otherwise powerful thalamocortical rhythms, analogous to difference between burst to tonic modes?
15. Results: ‘trending towards significance’ - delete’

Reviewer #4

(Remarks to the Author)

Version 1:

Reviewer comments:

Reviewer #1

(Remarks to the Author)

Thank you for this thorough revision. All my previous points have been fully addressed. Congratulations!

Reviewer #2

(Remarks to the Author)

The authors have responded to some of my concerns, but not all. As such, many of the original issues persist.

1) I pointed out in my original report that the paper is not adequately written, especially considering the broad readership they target. I felt that the results appear as a collection of findings that are not coherently presented into a story, and many of the analyses appeared ad-hoc. In their response, the authors said they have re-organized and focused the manuscript throughout to better justify and tie together examinations of stream-specific effects of CG feedback and effects on extraclassical surround suppression. However, they are not specific regarding what exactly they did and in fact, I don't feel that the manuscript and presentation are improved in a any significant manner.

2) I was originally puzzled by the low number of cells across the studies, in the 20-30 range across 3 animals. The main issue at play here is the robustness of their results and more cells would have definitely increased confidence. I also wanted to see the multiple comparison statistics being performed, such as Bonfferoni correction applied to their data. They performed multiple tests on the same population of neurons, so that analysis would be justified. I also asked how many of their cells exhibit statistically significant effects as I couldn't find this information anywhere in the manuscript.

In their response, the authors acknowledged the validity of my requests, but they made little effort to address them. For instance, they now summarize the total number of LGN recording penetrations performed, and the total number of well isolated single units they identified, along with “hit rates” for recording good LGN units across monkeys. This is good. However, it does not address the low cell count issue raised originally. Regarding statistical tests, they mention that the appropriate test for their comparisons is the paired t-test. This is incorrect, however, a nonparametric test, such as Wilcoxon signed-rank, would be more appropriate than their parametric test they employed (t-test). They are also claiming that they obtain “a single value per condition, and there is no reliable test for individual neuron significance”. However, this is incorrect, they have multiple trials for each neuron that was recorded, which should allow them to assess statistical significance on a cell basis, and then test for multiple comparisons. Instead of arguing why this is difficult, they could simply perform the appropriate analyses that I have requested in the first place.

3) I asked them to provide a justification for the many analyses on the data, and the need to employ a battery of stimuli to examine the role of CG feedback. However, no justification is provided. For instance, why spatial and temporal frequency, and why is temporal precision important in this context, and how is it related to the effects of surround suppression, and why is it important to study these effects across species? The authors' response was generic, i.e., saying that they addressed my concerns in the manuscript, but nothing specific was provided, and their paper didn't get better.

Reviewer #3

(Remarks to the Author)

The authors have addressed each of my points, and have provided appropriate changes or evidence in each case. The manuscript is improved in organisation and clarity, and I believe these are important observations. However I do think there could still be improvements to the figures and their descriptions as below.

Point 13. Fig 5 still lacks some clarity - I think there is an error in the labelling of the lines in the legend at the border of panels B/C (Green = No LED, Parvo, M-cone?), which may be throwing me off. However, I think arrows indicating timing peaks like that in Panel A would be useful, and the complete flip of the S-cone response in panel C kernel needs to be explained. In addition, the response latencies of magnocellular cells are longer than those of parvocellular cells in panel D - this seems counter to the standard expectation, and the reason for this could/should be explained to help the reader.

Point 3. I appreciate the addition of Fig 4E in response to my comments, however I had more been suggesting a description of the different components of the receptive field and the influence of CG feedback / opto modulation, than a depiction of the spatial sensitivity of the receptive field. Regardless, I am not sure it works as an addition to Fig 4 and it might be more useful as a standalone (e.g. as schematic endpoint to summarise for the non expert reader).

Point 10. The figures now omit animal-identification, on basis that tuning properties in each cell class were similar across animals and thus units could be safely combined. This seems reasonable, however I remain unsure whether treating all units from all the animals as a single group is the best approach, or whether the analyses could factor in the individual animal as well as the individual unit. The authors could transparently consider the potential limitations of their chosen statistical approach, by for example considering Fries and Maris (2022, Journal of Cognitive Neuroscience 34(7):1114-1118 'What to do if N is two?') in the 'Statistical analyses' section.

Reviewer #4

(Remarks to the Author)

Version 2:

Reviewer comments:

Reviewer #2

(Remarks to the Author)

The authors have addressed all my concerns.

Reviewer #3

(Remarks to the Author)

The authors have addressed all my comments. The confirmatory analyses within animal are heartening.

We thank all four reviewers for their careful assessment of our manuscript and for their overall enthusiasm for our study. We have significantly revised the manuscript in accordance with the reviewers' comments. Changes to the manuscript text are indicated by red text. Point-by-point responses to each individual Reviewer concern are listed below.

.....

Reviewer #1 (Remarks to the Author):

Mai et al very elegantly demonstrate in monkeys and ferrets the effect of V1 cortical feedback on LGN neuron receptive field properties. Previous research has relied on classical non-selective approaches, such as pharmacological inactivation or cooling of V1. Here, the application of cutting edge retrograde optogenetic techniques allows for a specific targeting of V1 cortico-geniculate (CG) neurons projecting as feedback to LGN. Applying optogenetic methods in primates has proven to be a major challenge and therefore the new findings provide important conceptual and methodological advancement. The authors findings move the field ahead towards much needed cell specific pathway mapping in primate circuits.

My only major concern relates to the specificity of the optogenetic technique and how it might have contributed to the specific magnocellular effects on LGN receptive field (RF) size and surround suppression. Previous research has demonstrated such effect also for the parvocellular system (Jones et al, J Neurosci, 2012). However, such effects might be smaller for parvo in comparison to magno (Alitto & Usrey, Neuron, 2008) and based on their timing inconsistent with CG feedback effects. I believe a more extensive demonstration and if possible, quantification of the opto construct expression would be desirable. I will try to make a few specific suggestions for the authors' consideration.

1. Figure 1Bi and Bii nicely show LGN expression in monkeys L and C. In comparison the expression appears much weaker in monkey K (Biii). Moreover the expression at least in the displayed sections seems primarily focused on parvocellular, and not so much magnocellular layers. With this expression pattern the absence of parvocellular RF size effects seem surprising. Could you please comment on the strength of expression magno vs parvocellular, and how it might affect the electrophysiological results?

Reviewer 1 Response 1: We appreciate this concern as stream-specific effects could be confounded by differential virus expression. Unfortunately, virus expression cannot be quantified from LGN sections because little ChR2/mCherry expression is expected within the LGN. Expression usually requires intracellular machinery and viruses can only infect neurons through axon terminals, so the only cells within the LGN to express mCherry are inhibitory interneurons (and these are rarely infected, perhaps due to their axodendritic processes – see handful of cells in Figure 1Bi). Some darker staining along injection tracks is observed in some (as in Figure 1Bi and Bii), but not all cases (as in Figure 1Biii). The source of this injection track staining is not known (perhaps antibodies bound mCherry expressed within residual viral particles that did not infect axons). As the Reviewer intuits in point 2 below, a more accurate measure of expression may be to inspect the relative proportion of virus-infected CG neurons in the upper versus lower portions of layer 6. Accordingly, we performed this analysis as a better quantification of expression across parvocellular and magnocellular-projecting CG neurons. For each monkey, we selected 10 unique sections from separate slides that included V1 regions near recording sites. We then selected 1 counting window per section and counted all virus-labeled CG neurons within a 5X objective view. All sampled counting windows included at least 5 virus-labeled CG neurons. We

counted the number of CG neurons in upper and lower layer 6 and computed the average percentage of lower layer 6 CG neurons (putative magnocellular-projecting CG neurons) per monkey. We observed 31-46% of virus-labeled CG neurons in lower layer 6 across monkeys, consistent with prior reports of more parvocellular- than magnocellular-projecting CG neurons (Briggs et al 2016, Lund et al 1975). The Methods (last paragraph) and Results (first paragraph) have been revised to describe the counting procedure and results of this analysis, respectively.

As a secondary measure of the relative amount of expression across NHPs, we also compared the magnitude of optogenetic (LED-only) activation of LGN neurons across NHPs. We used the fold-change in maximum (stimulus-evoked) firing rate with LED stimulation across 2-3 stimulus conditions for the 8 directly opto-tagged LGN neurons in our dataset (4 from Monkey K, 3 from Monkey C, 1 from Monkey L). Average firing rate fold-changes across NHPs were similar: Monkey K = 1.16 ± 0.15 , Monkey C = 1.18 ± 0.26 , Monkey L = 1.14 ± 0.06 . The Results (second paragraph) have been revised to indicate similarities in optogenetic activation across monkeys. In summary, both relative expression of virus-labeled CG neurons in the upper and lower portions of layer 6 and optogenetic activation of LGN neurons revealed similar patterns of expression across monkeys in our sample.

2. Briggs (Neuron, 2026) and Fitzpatrick (Vis Neurosci, 1994) report a bimodal expression of parvocellular vs magnocellular projecting CG neurons in layer 6 the upper and lower third of layer 6, respectively. From Figure 1C it seems that this is also the case here. Again, with this expression pattern the absence of parvocellular RF size effects seem surprising, and the authors may want to comment further.

Reviewer 1 Response 2: This is an excellent point and given that we demonstrate expression in both parvocellular- and magnocellular-projecting CG neurons (see Reviewer 1 Response 1 above), this strengthens our argument that stream-specific effects are not confounded by unequal expression levels. We have added to the Discussion (second paragraph) text highlighting this important point.

3. From Figure 1Bi expression in magnocellular neuron cell bodies seems a possibility. If this is the case this might open up the possibility of antidromic activation of LGN axons to V1, at least in theory. In my experience, such imperfections of virus expression are usually present, though rather minor, in comparison to the principal mechanism. Nevertheless, if antidromic activation remains a minor possibility, the authors may want to opt for more cautionary wording.

Reviewer 1 Response 3: Although “promiscuous” virus expression has been observed with many viruses, we have never observed this with G-deleted Rabies virus. Rabies is strictly retrograde and only infects cells whose axon terminals are near the injection site (Callaway 2009, Ghanem & Conzelmann 2016, Kelly & Strick 2000, Ugolini 2010, Wickersham et al 2007). We do observe virus-labeled neurons within the LGN (e.g. Figure 1Bi), but these are inhibitory interneurons, i.e. the only LGN cell type with axon (or axodendritic) terminals within the LGN. Virus-labeled LGN neurons in our sample have unique morphologies (different from relay cells) and lack dendritic spine-like processes, as previously described for LGN inhibitory interneurons (Guillery 1966, Hendrickson et al 1983). In our sample, interneurons are labeled less frequently than we might expect, given that inhibitory interneurons make up ~10% of LGN neurons (Sherman & Guillery 2006), perhaps because their axodendritic processes are unique and often these synapses are within glomeruli (Sherman & Guillery 2006). Furthermore, dense axonal label in layer 4C would be expected if LGN relay cells were infected with the virus. Only dendrites of CG neurons are

present in layer 4C (Figure 1Ci, Cii). We have revised the Results (first paragraph) to state that LGN relay cells and their axons in layer 4C are not labeled, preventing antidromic LED activation.

4. It would be good get more detail about the optogenetically identified CG neurons, including whether they are visually driven and to what extent their receptive field (RF) overlaps with the sampled LGN neurons. The degree of RF overlap is known to be a critical variable and excitatory vs inhibitory effects have been reported, at least for parvocellular neurons (Jones et al, J Neurosci, 2012).

Reviewer 1 Response 4: New Supplemental Figure 1 illustrates the overlap between receptive field centers of simultaneously recorded LGN and V1 neurons for all recording penetrations that yielded neuronal recordings used in this study. Pairs of recordings that included LGN (dark blue) and/or V1 neurons (light blue) that were directly modulated by the LED-alone (i.e. “opto-tagged” neurons) are illustrated separately from pairs of recordings that did not include opto-tagged neurons. Across all penetrations, separation between receptive field centers for recorded LGN and V1 neurons was small (average distance across all pairs of recordings= 0.47 ± 0.11). This information has been added to the Methods (Neurophysiological recordings section). It was perhaps not surprising that receptive field center separation for LGN/V1 recording pairs that included opto-tagged neurons was significantly smaller than separation among recording pairs that did not include opto-tagged neurons ($p=0.029$, rank sum test; separation for pairs with opto-tagged neurons= 0.17 ± 0.11 degrees, for pairs without opto-tagged neurons= 0.67 ± 0.15 degrees). This information has been added to the Results (second paragraph). Importantly, the largest separation distances between LGN and V1 receptive fields were less than 2 degrees (see new Figure S1), and the LED illuminated a larger region of cortex than the location of the V1 electrode. Furthermore, prior results in primates suggest that inhibitory effects of non-overlapping V1 feedback on LGN activity do not occur until there is at least 2 degrees of separation between LGN and V1 receptive fields (Jones et al 2012), which is larger than the largest separation distance among our penetrations. In their introduction and here, the Reviewer mentions Jones et al (2012) as showing effects of corticogeniculate feedback on the surrounds of parvocellular neurons. We would argue that this interpretation is not straightforward given the blunt method utilized (GABA-antagonist injections into V1 either aligned or mis-aligned to recorded LGN neurons). Instead, this paper shows a variety of gain changes in parvocellular neurons depending on the overlap between the injection site and the LGN recording area. The authors did not sample magnocellular neurons and did not compute extraclassical surround suppression in a rigorous manner. Therefore, it is difficult to interpret their findings beyond the fact that gain changes depend on location and spread of global V1 inactivation.

In response to the Reviewer’s question about tuning of CG neurons: most opto-tagged V1 neurons (8 of 12) displayed tuning along at least one stimulus dimension (e.g. tuned for stimulus orientation), with tuning preferences similar to those observed previously for antidromically identified CG neurons (Briggs & Usrey 2009). Given the small sample of putative CG neurons and the focus of this manuscript on LGN responses, we did not include tuning data for V1 neurons.

Minor:

5. I appreciate different reporting styles and do not want to criticize the authors. From my perspective, I appreciate explicit reports on negative effects in the abstract, such as the suspected lack of surround suppression in parvocellular neurons.

Reviewer 1 Response 5: The abstract has been revised as recommended.

6. For experiments that apply both visual as well as optogenetic stimulation, the authors refer to synchronization between LED and visual stimulation. Could you please provide additional details how this synchronization was achieved and maintained throughout varying stimulus conditions and cell classes?

Reviewer 1 Response 6: These details have been added to the Methods (Optogenetic stimulation section).

7. I would appreciate defining the m-sequence stimulus more clearly. Including a schematic graphic of the stimulus in different conditions (achromatic, cone-modulating separately) could be very helpful in comprehensive understanding of the experimental design, allowing for better visualization and interpretation of the stimulus effects.

Reviewer 1 Response 7: Schematics of achromatic (black/white) and cone-modulating m-sequence frames have been added to revised Figure 5.

8. Consider adding an indication of significance to Figure 3. The figure appears complex, with abundant information. The arrows on the plots might be misleading, as they point at both trends and significant differences. To enhance clarity, bar plots showing average results for each cell type could be included.

Reviewer 1 Response 8: Figure 3 has been revised as recommended by multiple reviewers to indicate significance effects more clearly. Bar plots showing significant differences across neuronal types (above scatter plots) and showing significant LED effects (right of scatter plots) are now included along with asterisks to indicate significant differences.

9. In Figures 4, 5D, and 6, there is also no clear indication of significance. Arrows are also present in plots where the difference is not significant or nonexistent (4C, 5D). Consider either:

- o Removing arrows in all plots where the difference is not significant.
- o Adding a star or other indicative symbol to highlight significant findings.

Reviewer 1 Response 9: Figures 4-6 (and Supplemental Figures 4 and 5) have been revised as recommended by multiple reviewers to indicate significant effects more clearly and to insure consistent statistical reporting across all scatter plots. Notably, arrows have been removed and significant differences are indicated on distributions with asterisks.

Reviewer #2 (Remarks to the Author):

This is the first study, to my knowledge, using optogenetic activation and inactivation of corticogeniculate feedback (CG, V1 to LGN) to examine its role in LGN information processing and coding. They discovered stream-specific effects of CG feedback in improving the spatial resolution of magnocellular LGN neurons, and found that enhancing CG feedback increased extraclassical surround suppression, shranked the size of LGN receptive fields, and increased preferred spatial frequencies of magnocellular LGN neurons. In turn, optogenetically suppressing CG feedback reduced surround suppression and increased preferred stimulus size. They further conducted cross-species comparisons (anesthetized monkeys and ferrets) to find that enhancing CG feedback reveals similar stream-specific effects in LGN Y neurons, and that optogenetically enhancing CG feedback improved LGN response reliability and reduced response jitter across neuronal types. Overall, this study is timely and reports some interesting findings although I'm unsure about the general relevance for the broad readership of Nature Communications. In particular, they seem to report results that are not necessarily connected. In addition, the cell count is very small, and they lack multiple comparison statistics. So I'm not fully convinced of the validity and robustness of the results, as presented here, and how exactly they will advance the field.

Reviewer 2 General response: We have reorganized the manuscript to enhance the focus and justify each of the experimental tests. Multiple comparisons corrections have been applied where necessary. More importantly though, specific statistical testing procedures have been clarified throughout.

Specific comments:

1) The paper is not well written, especially considering the broad readership they target. As presented, the results appear as a bunch of findings that are not coherently presented into a story, many of the analyses appear ad-hoc, and the figures are not effective or self explanatory. Also, the paper is not written in the NPG format, for instance Materials and Methods appear immediately after Introduction. The statistics are presented in a table, but I feel they should have been part of the main text or the figure captions. I found it puzzling when they present the scientific premise that they don't cite the relevant papers in the very field they examine. There have been several papers already manipulating cortical feedback optogenetically (although not CG feedback), but these studies are not mentioned. In particular, a relevant study from Angelucci's lab (Nurminen et al., Nat Commun, 2018), in which the authors manipulated cortical feedback from V2 to V1 to alter responses and surround suppression is not even mentioned. I think that study is very relevant since it is possible that some of the surround suppression results the authors report here could originate from V2.

Reviewer 2 Response 1: We appreciate this feedback; and we have re-organized and focused the manuscript throughout to better justify and tie together examinations of stream-specific effects of CG feedback and effects on extraclassical surround suppression. The Materials and Methods section has been moved to follow the Discussion, following the Nature Communications format. Significant comparisons are now noted in the text throughout the Results section. We prefer to leave most of the detailed statistics in the Tables to reduce lengthy and cumbersome parentheticals within the manuscript itself. We appreciate this is a matter of personal preference though and can adjust the formatting if that is required.

We thank the Reviewer for bringing the Nurminen et al (2018) paper to our attention as it is very applicable to our study. We now cite this paper at multiple places in the manuscript, including in the Introduction (fourth paragraph) and Discussion (sixth paragraph).

2) I am somewhat puzzled by the low number of cells across the studies, in the 20-30 range across 3 animals. True, they had many conditions they tested and controls, but they nonetheless used anesthetized animals and could have easily doubled the cell count. The main issue is the robustness of their results and more cells would have definitely increased confidence. In some of the analyses they used controls in which $n=4$, which is inappropriate. I would also like to see multiple comparison statistics, such as Bonferroni correction, applied to this data. They performed multiple tests on the same population of neurons, so this analysis is justified. Also, how many of their cells exhibit statistically significant effects? I couldn't find this information anywhere in the manuscript, but this is important knowledge that is currently missing.

Reviewer 2 Response 2: The Reviewer makes a good point that additional information is required to appreciate the cell counts for this study. In the Methods (Neuronal identification and tuning measurements section), we now summarize the total number of LGN recording penetrations performed (all using a 7-channel Eckhorn array), the maximum number of recordings that would be possible, and the total number of well isolated single units we identified, along with "hit rates" for recording good LGN units across monkeys (hit rate ~75% for all monkeys). We then clarified our further visual responsiveness/tuning criterion that reduced our sample to 98 visually responsive and tuned LGN neurons that were included in the manuscript.

We also clarified the statistical reporting throughout, as this was not clear in our previous submission. The Statistical analysis section of the Methods has been updated, as have the legends for the four tables. Importantly, most comparisons in the manuscript were comparisons of a single tuning metric (e.g. preferred spatial frequency) measured from one neuron across two conditions (with or without LED activation). The appropriate statistical test for this type of comparison is a paired t-test. In this case, with a single value per condition, there is no reliable test for individual neuron "significance". Instead, the paired t-test determines whether a population of similar neurons (e.g. LGN neurons of a defined cell type) demonstrates a systematic change in their responses across the two conditions. Because each population is unique and non-overlapping, there is no multiple-comparisons correction. However, there is one case in our manuscript where the same population was sampled in multiple ways: reliability and jitter comparisons were made sampling LGN cells up to four ways: all neurons, On or Off neurons, parvocellular or magnocellular neurons. For these tests, multiple comparisons corrections were applied, as described in the Methods (Statistical analysis section) and in the Table 3 legend.

In the last paragraph of the Results, we compared LED-induced gain changes in opto-tagged versus non-opto-tagged neurons and this was the only example in the manuscript where the n was low ($n=5-6$ in three instances). These data descriptions were included only for qualitative purposes, no statistical tests were performed on these subsets of data.

3) They performed many analyses on the data, and even used a battery of stimuli to examine the role of CG feedback, however no justification is provided for that. Why spatial and temporal frequency, and why is temporal precision important in this context, and how is it related to the effects of surround suppression, and why is it important to study these effects across species? Currently, these manipulations and analyses appear simply thrown out to see what happens when probing the impact of CG feedback, but the coherent story is missing.

Reviewer 2 Response 3: As described above, we have revised the manuscript throughout (Abstract, Introduction, Results, and Discussion) to justify the use of a comprehensive stimulus set designed to evoke different responses across LGN types (i.e. to test for stream-specific effects). We have also modified the manuscript throughout to tie together our examination of stream-specific effects of CG feedback with effects of CG feedback on surround suppression.

4) They conclude that the effects of CG feedback on the different metrics tested in LGN neurons are uncorrelated, but this is a bit surprising and the authors don't discuss the implications. Also, they present the magnocellular stream-specific effects of CG feedback on spatial receptive field properties as surprising, but this is actually expected. Overall, their results are not that surprising.

Reviewer 2 Response 4: The last two paragraphs of the Results have been revised for improved clarity. The second-to-last paragraph of the Results has been re-written to begin with a hypothesis that stream-specific effects should mean a lack of correlation between LED effects on spatial resolution and temporal precision. This is because spatial resolution effects are stream-specific and temporal precision effects are not. Set up this way, we now include a more complete discussion of these results and their implications.

It is not clear why magnocellular-specific effects of CG feedback on spatial resolution are expected. For example, Nurminen et al (2018) showed quite uniform effects of manipulating corticocortical feedback across V1 neurons (they did not report information about functional cell types). This would suggest that feedback influence over spatial resolution should generalize, and NOT be specific to one stream. A comparison of corticocortical and CG feedback in the context of spatial resolution has been added to the Discussion (sixth paragraph).

5) The figures are difficult to follow, and it is unclear whether they support the claims. They should clearly label individual examples and population effects, and indicate the statistical significance of the results, otherwise it is difficult to go back and forth to the text and the table to understand what's going on. They should also clearly label optogenetic activation and inactivation on each figure, not only in the text accompanying the figures.

Reviewer 2 Response 5: We apologize for the lack of clarity in our figures. We have significantly revised Figures 3-6 and Supplemental Figures 4 and 5 to clearly show statistically significant effects, distinguish example neurons from population results, and more clearly indicate conditions with and without LED stimulation.

Reviewer #3 (Remarks to the Author):

The observations here provide novel empirical data on a matter of strong interest – the impact of V1 projections on the activity of neurons in monkey LGN, the main visual input to V1. While this first feedback pathway in the visual system has been the subject of much previous work, in primates that work has had to rely on bulk inactivation of cortex by physical or chemical means. The current experiments use viral mediated optogenetic stimulation of cortical layer 6 neurons to provide much greater specificity. The measurements are technically challenging, requiring viral injection and precise electrophysiological measurements from a small and very deep thalamic brain structure, and successful optogenetic activation or suppression of neurons in cortex. An additional strength is that complementary measurements in ferret provide comparative data.

The main finding is that increasing cortical feedback changes spatial summation in Magno LGN pathway, alongside more general modulation of spike timing precision. The data is generally well described (though I think some of the aesthetics of the Figures could be improved), and the inferences are generally conservative. I therefore believe the experiments reported here are important addition to the literature.

Reviewer 3 Response 1: We greatly appreciate the reviewer's enthusiasm for our work. As recommended by all reviewers, we have significantly revised Figures 3-6 (and Supplemental Figures 4 and 5) to show significant effects more clearly.

I have some queries about particular aspects of the experiments and conclusions, below, but the main thought I had after reading was need for a clearer explanation of optogenetic effects on response amplitude. There are some rather large differences in the amplitudes of mean activity in some of the panels in Figure 2 – which I think are by and large not commented on, with focus on tuning. Better explanation of the impact of CG stimulation, for example, scatter plots of response amplitude in the LED / control conditions, might help the reader understand what is going on, but regardless the reader needs help here.

Reviewer 3 Response 2: The Results (fifth paragraph and last paragraph) have been revised to discuss in more detail optogenetic effects on LGN response amplitude/gain. Also, new Supplemental Figure 3 has been added which shows scatter plots of gain values, computed from area under the spatial and temporal frequency tuning curves, for all LGN neurons, as recommended. Statistics for all firing rate and amplitude/gain values across neuronal types and LED stimulation conditions are also listed in Table 2. Briefly, there were no significant effects of LED activation of CG feedback on measures of LGN neuronal response amplitude from tuning curves. We did observe a significant optogenetic facilitation of amplitude of the STA from white noise stimuli for magnocellular neurons only. Also, there were some significant relationships between firing rate and response gain whereby a subset of LGN neurons, presumably those receiving convergent input from ChR2-expressing CG neurons, showed more consistent LED-mediated effects. All of these observations are now more clearly laid out in the Results.

On a related note, while there appear to be limited effects of optogenetic stimulation on average 'spontaneous' activity (Table 2), it is possible that optogenetic stimulation entrained activity without elevating or reducing it – it would be useful to see whether there was temporal modulation of firing rate at the rates of LED stimulation, in spontaneous or visually driven cases, where that is possible.

*Reviewer 3 Response 3: Following Reviewer 3's suggestion, we performed power spectral analyses of LGN and V1 spiking and local field potentials (LFPs) to test whether LED stimulation entrained LGN or cortical activity and to test whether these effects differed during spontaneous activity or during visual stimulation. Power spectra from both LGN and V1 spike trains recorded during trials in which the LED was pulsed at varying frequencies (in the absence of any visual stimulus) did show peaks corresponding to LED pulse frequencies (e.g. 4Hz, ~14Hz, ~26Hz, and their harmonics; **Response Figure 1A**, power spectra from spiking activity shown but LFP spectra were similar). These findings are consistent with our observed opto-tagged responses among LGN and V1 neurons.*

*Interestingly, when we examined power spectra (from LGN and V1 spikes and LFPs) from trials with visual stimulation alone and trials with visual stimulation synchronized with LED activation, we did not see any differences in power across frequencies. This lack of difference was replicated across all three monkeys (**Response Figure 1B**, power spectra from LFPs shown, but spiking activity spectra were similar). Together these findings suggest that while LED stimulation did entrain some rhythmic spiking activity, visual stimulation "overpowered" this rhythmic activity, consistent with the notion that LED stimulation in our experimental protocol was small relative to visual stimulation. We have described these observations in the Results (last paragraph).*

Response Figure 1: Power spectral analyses of LGN and V1 spikes and LFPs. **A.** Power spectra from spiking activity in LGN (blue) and V1 (cyan) on LED-only stimulation trials, in the absence of

visual stimulation. Shaded regions are SEMs. Data combined from 3 LGN (up to 7 contacts each) and 3 V1 recordings (up to 24 contacts each) per monkey. Note peaks ~4Hz, etc. **B.** Power spectra from LFPs in LGN to visual stimulation alone (black) and with LED (blue) stimulation, and in V1 to visual stimulation alone (yellow) and with LED (cyan) stimulation for each monkey (3 penetrations each, number of contacts as above). Shaded regions are SEMs. Small differences are evident between LGN and V1 LFPs, but not across LED conditions per area.

Inclusion of a descriptive but quantitative model (not one aimed at explaining the specific observations) may also help make the conclusions more coherent. In early parts of the Results you state that “Moreover, these effects are probably linked: by increasing extraclassical surround suppression, the classical receptive field area, and thus preferred stimulus size, shrink leading to an increase in the preferred grating spatial frequency. Thus, CG feedback is functionally stream-specific in the spatial domain, regulating receptive field size through extraclassical suppression for the magnocellular stream selectively”. This mechanism would also explain the later results where there is increased gain but no change in receptive field size in the M-sequence experiments (the classical receptive field is unaffected, and a change in gain in the M-sequence experiments could be explained by reduced extraclassical suppression driven by the spatially extended flickering stimulus). If the extraclassical receptive field, including the cortical-driven component, is larger and more sensitive to lower spatial frequencies (which is likely) than is the classical receptive field, this would likely provide a sufficient explanation of the variety of spatial effects of stimulation.

Reviewer 3 Response 4: As recommended, a schematic representation of the spatial resolution effects is now illustrated in new Figure 4E. Also, the Reviewer’s explanation for increased STA amplitude with LED activation of feedback for magnocellular neurons is an intriguing one, especially as the Reviewer is correct in that we expect extraclassical suppression to be minimally engaged by the white noise stimulus.

1. Introduction: Paragraph 2 spends a good deal of time reiterating fairly standard description of roles of different thalamus groups (Parvo, Magno, Konio) however it feels unneeded here; would be good to focus this paragraph onto what the reader needs to know about CG feedback in particular.

Reviewer 3 Response 5: Paragraph two of the Introduction has been shortened, but we feel it is important to keep the details of the feedforward streams as well as the physiology of the CG types in this paragraph. We often get comments that many readers are not familiar with (or don’t remember) the physiological particulars of the parallel streams in primates. We have refocused the paragraph onto relationships to the CG pathways, as recommended.

2. Introduction: Paragraph 3 notes several different studies in various species on impact of causal manipulations of CG feedback. However the techniques used vary immensely and are many were fairly crude (no disrespect) disruptions, as is detailed in Discussion - it would be useful the reader is aware of that here (and thus contrasting the utility of the optogenetic approach).

Reviewer 3 Response 6: The third paragraph of the Introduction has been revised as recommended.

3. Introduction: Paragraph 3: there is enough debate about the link between X/P and Y/M (as you make clearer in Discussion) that I am not sure it is necessary to be strong on the link here - just need to note that there are different pathways and different nomenclatures for different species.

Reviewer 3 Response 7: The third paragraph of the Introduction has been revised as recommended.

4. Introduction: Paragraph 4: its not clear why visual stimuli that evoke unique responses across the parallel streams are important for understanding CG feedback. I may be missing something here, but its not spelled out.

Reviewer 3 Response 8: Reviewer 2 raised an overlapping concern about tying together the stream-specificity story with the surround suppression story. Accordingly, the Introduction (especially fourth and fifth paragraphs) have been revised, including justifying the need for using a comprehensive set of stimuli designed to differentially modulate the activity of LGN neurons in each stream. The Results have also been revised throughout to emphasize the need for stimuli designed to evoke distinguishable responses across LGN neuronal types in the pursuit of understanding whether CG feedback is functionally stream-specific.

5. Methods: 'Surgical procedures...' I think that both sufentanil and isoflurane were provided throughout the recordings, at level depending on physiological indicators, but would be good to make that explicit.

Reviewer 3 Response 9: The "Surgical procedures..." section of the Methods has been updated to include details of anesthetics used. This section also includes a statement about maintaining adequate anesthesia levels according to physiological indicators.

6. Methods: 'Optogenetic stimulation'. What was the rationale for adopting the three methods of stimulation? I think I can discern, but would be good to spell out. I am also remain somewhat unsure when each protocol was used, and would like to see this made clearer.

Reviewer 3 Response 10: The "Optogenetic stimulation" section of the Methods has been revised to include description of which protocols were used per monkey and the rationale for adding more protocols (increase the likelihood of detecting opto-tagged neurons). A second revised paragraph in this same section more clearly describes the synchronization of the LED with visual stimuli.

7. Methods: 'Visual stimulation'. "in steps of 10" -> "in 10 steps"?

Reviewer 3 Response 11: Edited as recommended.

8. Methods: 'Visual stimulation': How were the flashing spots positioned relative to receptive fields?

Reviewer 3 Response 12: We apologize that this section was unclear. All stimuli (gratings, spots, m-sequence grids) were centered over the receptive fields of recorded neurons. In many cases, multiple neuronal receptive fields were proximal and overlapping – in this case a single stimulus

could activate all receptive fields. In other cases where receptive fields were more spatially displaced, multiple stimuli (gratings or spots) were positioned at the centers of each corresponding receptive field. The “Visual stimulation” section of the Methods has been revised to make this clearer.

9. Methods: ‘Neural identification and tuning measurements’: Power law for contrast analyses: As I think a power law with exponent 2 only allows expansive relationships there is no saturation in contrast-response curves and $c50$ is largely undefined, so I am unsure how it is obtained. Also some of the fits in Figure 2, left column, look compressive not expansive. Am I missing something here?

Reviewer 3 Response 13: As recommended, we used our original $c50$ s computed from Naka Rushton fits, rather than using those from power law fits. Importantly, $c50$ values did not change by more than 2% for any neuron, so overall results were unchanged. The text of the Methods section on contrast curve fits has been revised to only describe the Naka Rushton fits.

10. Methods: ‘Statistical analyses’ Optogenetic activation is used in 3 hemispheres of 3 animals, and suppression used in 1 hemisphere of one animal. The low number of animals is to be expected in monkey experiments, and the authors have been able to obtain good numbers of LGN neurons in each animal – as in usual practice, most of the statistics pool observations across animals to investigate impact on individual neurons. However, a likely source of variance across neurons is the extent / expression of viral product in each animal, so it would be good to provide additional analyses which take animal into account. This may be particularly important in analyses where individual animals provide substantial majority of data (e.g. latency to flashing spots; Figure 6).

Reviewer 3 Response 14: In response to this and concerns raised by Reviewer 1, we performed new anatomical and neurophysiological analyses to demonstrate similar levels and patterns of virus expression and optogenetic effects across all three NHPs (see Reviewer 1 Response 1). Additionally, revised Supplemental Figure 2 illustrates multi-dimensional tuning across LGN neurons per type and also indicates with symbols which NHP the neurons were recorded in. This plot also shows that neurons of the same type recorded in different NHPs showed remarkably similar multi-dimensional tuning. Given this demonstration of similarity, we then removed all animal-specific symbols from Figures 3-6 to improve the clarity and interpretability of these figures.

11. Results: it would be useful if Figure 1 or similar could show the overlap in receptive field locations of LGN and cortical neurons. As it stands, the only information I can find on the alignment of cortical stimulation/recording location, and LGN recording location, is the first sentence of paragraph 2 in results.

Reviewer 3 Response 15: We now provide new Supplemental Figure 1, which illustrates the center-to-center distances between receptive fields for simultaneous LGN and V1 recordings. In the “Neurophysiological recording” section of the Methods, we also report the average center-to-center receptive field distances for these recordings. See also Reviewer 1 Response 4 for additional details.

12. Results: there seems to be a missed chance to provide a more detailed description of the 8

LGN neurons directly responsive to cortical optogenetic activation. For example, were there any functional properties that might suggest particular class (e.g. interneurons, or Parvo/Magno/Konio).

Reviewer 3 Response 16: Of the 8 opto-tagged LGN neurons, 4 were parvocellular neurons, 3 were magnocellular neurons, and 1 was a koniocellular neuron. All had significant tuning for multiple visual stimulus tests (e.g. contrast, SF, TF). One (a parvocellular neuron) had a large increase in maximum visually evoked firing rate and large gain increases with LED activation, but the others did not show consistent increases in maximum visually evoked firing rate or response gain. Since there were no discernable relationships between physiological response properties and LED modulation among opto-tagged LGN neurons, we did not include further information on them in the text. However, we did revise the last paragraph of the Results to better describe firing rate and amplitude changes across our sample of LGN neurons, and possible links to opto-tagged neuronal responses.

13. Results: Figure 5 – a fairly strong text statement about ‘many LGN neurons had shorter spiking response latencies’ and ‘note blue curves are shifted towards 0’ in Results is at odds with the modest and even reversed timings in the Figure. Some clarification would be useful.

Reviewer 3 Response 17: We apologize that this (tenth) paragraph of the Results corresponding to Figure 5 was not clear. This paragraph has been revised to explain that across the population of LGN neurons, response latency computed from white noise STAs was in fact significantly reduced with LED activation of CG feedback (as shown in Figure 5D, stats reported in the text).

14. Results: it may be unlikely, but did optogenetic stimulation have an impact on cortical or LGN LFP power spectrum? That is, could reduced spike timing variability in LGN be related to disruption of otherwise powerful thalamocortical rhythms, analogous to difference between burst to tonic modes?

Reviewer 3 Response 18: This is an interesting idea. We examined power spectra of LGN and V1 LFPs during visual stimulation without and with LED activation of CG feedback (see Response Figure 1 above). While power spectra from LGN and V1 LFPs showed some very subtle differences, there were no differences in power spectra across LED conditions (Response Figure 1B). This is consistent with the notion that the LED was not powerful enough to entrain new thalamocortical rhythms – these were still dictated by visual stimulus drive. Thus, effects on spike timing variability were likely due to local circuit and synaptic mechanisms rather than more global rhythms. We mentioned the results of the power spectral analyses in the last paragraph of the Results.

15. Results: ‘trending towards significance’ - delete’

Reviewer 3 Response 19: The two phrases mentioning trends toward significance have been removed.

Reviewer #4 (Remarks to the Author):

Reviewer 4 Response: We thank you for taking the time to review our work and appreciate your input in providing us with constructive feedback.

References

- Briggs F, Kiley CW, Callaway EM, Usrey WM. 2016. Morphological substrates for parallel streams of corticogeniculate feedback originating in both V1 and V2 of the macaque monkey. *Neuron* 90: 388-99
- Briggs F, Usrey WM. 2009. Parallel processing in the corticogeniculate pathway of the macaque monkey. *Neuron* 62: 135-46
- Callaway EM. 2009. Transneuronal circuit tracing with neurotropic viruses. *Current Opinion in Neurobiology* 18: 1-7
- Ghanem A, Conzelmann KK. 2016. G gene-deficient single-round rabies viruses for neuronal circuit analysis. *Virus Research* 216: 41-54
- Guillery RW. 1966. A study of Golgi preparations from the dorsal lateral geniculate nucleus of the adult cat. *Journal of Comparative Neurology* 128: 21-50
- Hendrickson AE, Ogren MP, Vaughn JE, Barber RP, Wu JY. 1983. Light and electron microscope immunocytochemical localization of glutamic acid decarboxylase in monkey geniculate complex: evidence for gabaergic neurons and synapses. *J. Neuroscience* 3: 1245-62
- Jones HE, Andolina IM, Ahmed B, Shipp S, Clements JTC, et al. 2012. Differential feedback modulation of center and surround mechanisms in parvocellular cells in the visual thalamus. *J. Neuroscience* 32: 15946-51
- Kelly RM, Strick PL. 2000. Rabies as a transneuronal tracer of circuits in the central nervous system. *Journal of Neuroscience Methods* 103: 63-71
- Lund JS, Lund RD, Hendrickson AE, Bunt AH, Fuchs AF. 1975. The origin of efferent pathways from the primary visual cortex, area 17, of the macaque monkey as shown by retrograde transport of horseradish peroxidase. *J. Comp. Neurol.* 164: 287:303
- Nurminen L, Merlin S, Bijanzadeh M, Federer F, Angelucci A. 2018. Top-down feedback controls spatial summation and response amplitude in primate visual cortex. *Nature Communications* 9: 1-13
- Sherman SM, Guillery RW. 2006. *Exploring the thalamus and its role in cortical function*. Boston: MIT Press.
- Ugolini G. 2010. Advances in viral transneuronal tracing. *Journal of Neuroscience Methods* 194: 2-20
- Wickersham IR, Finke S, Conzelmann KK, Callaway EM. 2007. Retrograde neuronal tracing with a deletion-mutant rabies virus. *Nature Methods* 4: 47-49

Response to Reviewer Concerns Round 2 – NCOMMS-24-45398-T

We thank all four reviewers again for their careful assessment of our manuscript. We greatly appreciate that the reviewers were pleased with our revisions from the prior round. We have made the additional revisions requested by Reviewers 2 and 3. Changes to the manuscript text are indicated by red text. Point-by-point responses to individual Reviewer concern are listed below.

.....

Reviewer #1 (Remarks to the Author):

Thank you for this thorough revision. All my previous points have been fully addressed. Congratulations!

Thank you for your enthusiasm and support for our work!

Reviewer #2 (Remarks to the Author):

The authors have responded to some of my concerns, but not all. As such, many of the original issues persist.

1) I pointed out in my original report that the paper is not adequately written, especially considering the broad readership they target. I felt that the results appear as a collection of findings that are not coherently presented into a story, and many of the analyses appeared ad-hoc. In their response, the authors said they have re-organized and focused the manuscript throughout to better justify and tie together examinations of stream-specific effects of CG feedback and effects on extraclassical surround suppression. However, they are not specific regarding what exactly they did and in fact, I don't feel that the manuscript and presentation are improved in a any significant manner.

Reviewer 2 Response 1: We apologize that we didn't describe the changes made to the text in the previous round in sufficient detail. In our prior revision, we revised the Abstract, Introduction, and Discussion to explain that in order to examine stream-specific effects of CG feedback, we needed to first define neurons as belonging to one of the three parallel streams. This requires a large battery of stimuli, including those classically used to define magnocellular, parvocellular, and koniocellular LGN neurons (e.g. gratings varying in contrast, spatial frequency, temporal frequency, and size). The second paragraph of the Introduction describes the differential preferences of macaque LGN cell types for stimuli varying in these parameters. Having defined LGN neurons based on their physiological responses, we then examined how LED activation of CG feedback alters their response properties, including responses to stimuli varying in size. In this revision, we added additional text to the Results to further clarify the use of stimuli to define cell types in order to assess whether CG feedback has stream-specific effects on LGN neurons.

2) I was originally puzzled by the low number of cells across the studies, in the 20-30 range across 3 animals. The main issue at play here is the robustness of their results and more cells would have definitely increased confidence. I also wanted to see the multiple comparison statistics being performed, such as Bonfferoni correction applied to their data. They performed multiple tests on the same population of neurons, so that analysis would be justified. I also asked how many of their cells exhibit statistically significant effects as I couldn't find this

information anywhere in the manuscript.

In their response, the authors acknowledged the validity of my requests, but they made little effort to address them. For instance, they now summarize the total number of LGN recording penetrations performed, and the total number of well isolated single units they identified, along with “hit rates” for recording good LGN units across monkeys. This is good. However, it does not address the low cell count issue raised originally. Regarding statistical tests, they mention that the appropriate test for their comparisons is the paired t-test. This is incorrect, however, a nonparametric test, such as Wilcoxon signed-rank, would be more appropriate than their parametric test they employed (t-test). They are also claiming that they obtain “a single value per condition, and there is no reliable test for individual neuron significance”. However, this is incorrect, they have multiple trials for each neuron that was recorded, which should allow them to assess statistical significance on a cell basis, and then test for multiple comparisons. Instead of arguing why this is difficult, they could simply perform the appropriate analyses that I have requested in the first place.

Reviewer 2 Response 2: Reviewer 2 is correct in that we should have described tests for distribution normality and included non-parametric tests (Wilcoxon signed-rank tests) where appropriate. We apologize that we did not include this in the prior round. The Methods and all Table legends have been updated to reflect use of paired t-tests for normally distributed samples and supplemented with Wilcoxon tests when necessary. We also have added Bonferroni correction for all tests across the same samples of neurons, as requested (revisions to Methods and Table legends). Importantly, these added measures did not change any of our overall findings. One comparison of TFhigh50 across LED conditions among ferret Y neurons was no longer significant after multiple comparisons correction (Supplemental Table 1) and the Results text describing this finding has been revised accordingly.

Regarding cell counts and overall statistical rigor: To provide some context for the sample size in our study, we examined sample sizes across 12 similar types of studies spanning 40 years (1984 to present including our study). All studies involved recording LGN neuronal physiology in anesthetized macaques. We also included in this analysis sample size data from Nurminen et al 2018 Nature Communications, as Reviewer 2 noted that this particular study was “very relevant” to our current study. The average number of LGN neurons per animal reported across studies was 24.7 (range: 4-79). We report an average of 32.7 LGN neurons per animal. In fact, our study has the third highest sample size from this group of studies. Furthermore, three of the studies involved manipulation of V1 during LGN recordings (cortical cooling or pharmacological agent injection), which required additional efforts to align LGN recordings with V1 manipulation sites (similar alignment efforts were required in our study). The average sample size for these three studies (not including ours) was 7.2 LGN neurons/animal. As a reference, Nurminen et al had a sample size of 11 cortical neurons per animal. Within this context, our sample size is substantially higher than that of prior studies involving V1 manipulation, and in fact is higher than that reported for the majority of anesthetized macaque LGN physiology papers over the past four decades. Our relatively large sample size therefore supports higher statistical rigor than is the norm in the field.

An additional note on multiple-comparisons testing and rigor: Application of multiple-comparisons corrections when a variety of independent tests (i.e. responses to different stimuli) are conducted on the same neuronal sample is inconsistent in the literature. For example, in the optogenetic manipulation study by Nurminen et al 2018, nine paired statistical comparisons across laser conditions were made from the same 33 neurons without any correction for multiple comparisons. How best to apply multiple comparisons testing on neuronal samples for which

many independent measures have been made is further complicated by the fact that many experimenters return to the same datasets in future studies, performing additional statistical comparisons. This makes it impractical to predict an appropriate multiple comparisons correction. We have applied corrections as recommended for all comparisons made among identical sets of neurons. We believe this to be rigorous and indeed higher than the standard set by recent studies.

Regarding individual cell significance: We appreciate that many of the LED effects we report here are small shifts in neuronal tuning preferences. Perhaps this is what is driving the Reviewer's request for tests of individual neuronal significance. Critically, for all comparisons of tuning metrics across LED conditions, we show values for every individual neuron (e.g. Figures 3, 4, and 7). For example, even though the increase in preferred spatial frequency is very small, Figure 3B shows that almost every black dot (magnocellular neuron) is above the diagonal. In other words, even though each individual magnocellular neuron shifted its preferred spatial frequency up by a small amount, what is significant here is the fact that almost ALL magnocellular neurons shifted toward higher spatial frequency preferences with LED activation of CG feedback. Similar consistent and small shifts are observed for magnocellular neurons in their preferred size responses and in their SSIs (Figure 3E-G).

The reviewer is correct in that we had ~30 trials per neuron per LED condition, but these trials used different stimuli (i.e. stimuli varying in a tuning parameter in 10 steps). I.e. we have only 3 repeats of any individual stimulus per LED conditions. This is an insufficient number of trials for within-neuron comparison. Also, it may be important to note that the tuning metrics we compare across LED conditions are not firing rates, but single value tuning metrics computed from tuning curves that are fit to all 30 data points per neuron per condition. An example is the surround suppression index (SSI). SSI is computed from the curve fit to neuronal responses to gratings varying in size (30 total trials, 3 trials each for 10 sizes). We obtain one SSI value for the no-LED trials curve and another for the with-LED trials curve. There is no statistical test to compare two values.

It may also be worthwhile to note that we did not observe consistent or significant LED effects on neuronal firing rates. So, all of the significant effects we observed were on neuronal tuning metrics (i.e. tuning preferences) and spiking reliability/variability measures, independent of significant changes in firing rate.

3) I asked them to provide a justification for the many analyses on the data, and the need to employ a battery of stimuli to examine the role of CG feedback. However, no justification is provided. For instance, why spatial and temporal frequency, and why is temporal precision important in this context, and how is it related to the effects of surround suppression, and why is it important to study these effects across species? The authors' response was generic, i.e., saying that they addressed my concerns in the manuscript, but nothing specific was provided, and their paper didn't get better.

Reviewer 2 Response 3: In our initial submission, we included more detailed descriptions of magnocellular, parvocellular, and koniocellular LGN neurons and how these cell types are defined by their differential responses to contrast, spatial and temporal frequency, and varying stimulus size. All Reviewers requested that we streamline the Introduction and reduce text devoted to defining the cell types in the parallel streams. The second paragraph of the Introduction still provides details about macaque LGN neuronal physiology, albeit streamlined. See Reviewer 2 Response 1 for additional explanation.

Reviewer #3 (Remarks to the Author):

The authors have addressed each of my points, and have provided appropriate changes or evidence in each case. The manuscript is improved in organisation and clarity, and I believe these are important observations. However I do think there could still be improvements to the figures and their descriptions as below.

Point 13. Fig 5 still lacks some clarity - I think there is an error in the labelling of the lines in the legend at the border of panels B/C (Green = No LED, Parvo, M-cone?), which may be throwing me off. However, I think arrows indicating timing peaks like that in Panel A would be useful, and the complete flip of the S-cone response in panel C kernel needs to be explained. In addition, the response latencies of magnocellular cells are longer than those of parvocellular cells in panel D - this seems counter to the standard expectation, and the reason for this could/should be explained to help the reader.

Reviewer 3 Response 1: We thank Reviewer 3 for catching some legend errors and an incorrect curve in this figure. We have revised (newly named) Figure 6B and C to include complete legends and we have used dashed blue lines with colored fills to indicate the with-LED responses per cone activation. The S cone response without LED stimulation in Figure 6C was incorrect in the previous figure (it was the Off response rather than the correct S-On response). We have added the correct curve to the revised figure. We have added arrows in B and C as requested.

Panel D gave an inaccurate representation of the magnocellular neurons' latencies because the red dots were on top of the black dots. We have rectified this. Importantly, there were no significant differences in response latencies measured from m-sequence pixel brightness curves between parvocellular and magnocellular neurons. See also Table 3 which shows very similar response latencies measured from the flashed spot stimulus among magnocellular and parvocellular neurons.

Point 3. I appreciate the addition of Fig 4E in response to my comments, however I had more been suggesting a description of the different components of the receptive field and the influence of CG feedback / opto modulation, than a depiction of the spatial sensitivity of the receptive field. Regardless, I am not sure it works as an addition to Fig 4 and it might be more useful as a standalone (e.g. as schematic endpoint to summarise for the non expert reader).

Reviewer 3 Response 2: We have removed this panel from Figure 4 and added it as a stand-alone (new) Figure 5. It is difficult in this experiment to tease apart whether CG feedback differentially modulates the classical center of the receptive field versus the extraclassical surround. Our metrics indicate a coordinated increase in surround suppression along with a shrinking of the classical receptive field. But we hesitate to speculate further on whether CG feedback may separately influence these receptive field components.

Point 10. The figures now omit animal-identification, on basis that tuning properties in each cell class were similar across animals and thus units could be safely combined. This seems reasonable, however I remain unsure whether treating all units from all the animals as a single group is the best approach, or whether the analyses could factor in the individual animal as well as the individual unit. The authors could transparently consider the potential limitations of their chosen statistical approach, by for example considering Fries and Maris (2022, Journal of

Cognitive Neuroscience 34(7):1114-1118 'What to do if N is two?') in the 'Statistical analyses' section.

Reviewer 3 Response 3: We appreciate that combining data across animals is a somewhat thorny issue in the field. We have added text describing how the main significant effects we observed (on surround suppression and response timing/precision) were significant in all three or at least two monkeys individually. We prefer not to add additional text beyond this as a broader discussion of animal numbers and combining data is beyond the scope of the current study.

Reviewer #4 (Remarks to the Author):

We thank Reviewer 4 for their thoughtful contributions.